# Evaluation of total ozone measurements from Geostationary Environmental Monitoring Satellite (GEMS)

Kanghyun Baek[1], Jae Hwan Kim[1], Juseon Bak[2], David P. Haffner[3], Mina Kang[4], Hyunkee Hong[5]

[1] Department of Atmospheric Science, Pusan National University, Busan, Republic of Korea
[2] Institute of Environmental Studies, Pusan National University, Busan, Republic of Korea
[3] NASA Goddard Space Flight Center, Greenbelt, MD 20771, USA
[4] Department of Atmospheric Science and Engineering, Ewha Woman's University, Seoul, Republic of Korea.
[5] National Institute of Environmental Research, Incheon, Republic of Korea.

*Correspondence to*: Jae Hwan Kim (jaekim@pusan.ac.kr)

**Abstract.** The continued interest in air pollution and stratospheric ozone variability has motivated the development of a geostationary environmental monitoring satellite (GEMS) for hourly ozone monitoring. This paper provides the atmospheric science community with the world's first assessment of GEMS total column ozone (TCO) retrieval performance and diurnal ozone variation. The algorithm used for GEMS is a more advanced version of its predecessor, the TOMS-V8, that incorporates several improvements, including a new look-up table, a simple Lambert equivalent reflectivity model, and a spectral dependence correction. The GEMS algorithm also uses the optimal estimation method (OEM) to make error analysis more accessible and robust. The estimated retrieval errors range from 1.5 to 2 DU in September and 2 DU in December, with a constant Degree of freedom of the signal (DFS) of 1 in September and a variable DFS of 1.25 to 1.4 in December throughout the day, depending on SZA. To assess the performance of the GEMS algorithm, the hourly GEMS total ozone was compared with ground-based measurements from Pandora instruments and other satellite platforms from TROPOMI and OMPS. GEMS has a high correlation of 0.97 and small RMSE values compared to Pandora TCO at Busan and Seoul. It is notable that despite exhibiting seasonal dependence in the mean bias of GEMS with Pandora, GEMS is capable of observing daily variations in ozone that are highly consistent with Pandora measurements, with a bias of approximately 1%. The comparison of GEMS TCO data with TROPOMI and OMPS TCO data shows a high correlation of 0.99 and low RMSE compared to TROPOMI and OMPS TCO data, but has a negative bias of -2.38% and -2.17% with standard deviations of 1.33% and 1.57%, respectively. Similar to OMPS, the influence of $SO_2$ from volcanic eruptions is not properly removed in some regions, leading to GEMS overestimating TCO in those areas. The mean biases of GEMS TCO data with TROPOMI and OMPS TCO are within ± 1% at low latitudes but become negative at mid-latitudes with an increasingly negative dependence on latitude. Furthermore, this dependence becomes more prominent from summer to winter. The empirical correction applied to the GEMS irradiance data improves the dependence of mean bias on season and latitude, but a consistent bias still remains, and a marginal positive trend was observed in December. Therefore, further investigation into correction methods is needed. The results are a meaningful scientific advance by providing the first validated, hourly UV ozone retrievals from a satellite in geostationary orbit. This

experience can be used to advance research with future geostationary environmental satellite missions, including incoming TEMPO and Sentinel-4.

## 1 Introduction

Stratospheric ozone is responsible for absorbing the sun ultraviolet (UV) radiation, protecting the Earth's surface from harmful UV rays. Ozone in the troposphere is a toxic air pollutant that affects human health via harmful respiratory and cardiovascular effects, and negatively affects vegetation growth (Crutzen, 1979; Jacob et al., 1999). Global ozone monitoring is therefore essential for both public health and environmental protection to provide valuable information about the state of the atmosphere and identify areas where public action is needed to reduce impacts on human health and the environment (Engel et al., 2018; Scientific Assessment of Ozone Depletion, 2014).

Satellite remote sensing is a powerful tool for monitoring atmospheric ozone with high spatial and temporal coverage of global observations (Fishman et al., 2008; Fishman and Larsen, 1987). Global ozone monitoring by Total Ozone Monitoring Spectrometer (TOMS) aboard the Nimbus-7 satellite in 1978 was the first mission dedicated to creating detailed maps of atmospheric ozone from space (Bhartia et al., 1996). Since then, the Global Ozone Monitoring Experiment (GOME) (ESA 1995), SCanning Imaging Absorption spectroMeter for Atmospheric CHartographY (SCIAMACHY) (Bovensmann et al., 1999), Ozone Monitoring Instrument (OMI) (Levelt et al., 2006), Ozone Mapping and Profiler Suite Nadir Mapper (OMPS) (Flynn et al., 2014), and TROPOspheric Monitoring Instrument (TROPOMI) (Veefkind et al., 2012), which all built on the success of TOMS, have provided a continuous and consistent mapping of atmospheric ozone.

The continued interest in air pollution and stratospheric ozone variability has motivated the development of new satellite missions with improved capabilities for hourly monitoring of atmospheric composition. The geostationary Air Quality (Geo-AQ) constellation missions such as Geostationary Environmental Monitoring Satellite (GEMS), Sentinel-4, and Tropospheric Emissions: Monitoring of Pollution (TEMPO) were designed to provide high-quality measurements of atmospheric composition throughout the day from geostationary orbit (Ingmann et al., 2012; Zoogman et al., 2017; Kim et al., 2020). These new missions will provide more accurate and timely information about air quality and stratospheric ozone for supporting air quality forecasts and policy making. The Geo-Kompsat-2B satellite carrying the GEMS sensor was the first mission of the Geo-AQ constellation, which was launched on 18 February 2020 (Kim et al., 2020). GEMS is a UV–visible spectrometer that measures direct solar irradiance and radiance backscattered from the Earth's surface and atmosphere covering the Asia-Pacific region (Kim et al., 2020).

Since GEMS is the first Geo-AQ mission, it is necessary to introduce the algorithm process and new data products to provide information for users. Here we focus on the GEMS total ozone (O3T) algorithm for retrieving total column ozone (TCO) from GEMS Level-1B (L1B) radiance spectra, and validation of these data using ground-based Pandora TCO measurements, and other satellite TCO measurements from OMPS and TROPOMI. The GEMS-O3T algorithm has several improvements over previous algorithms, such as the use of a new look-up table (LUT), simple Lambert equivalent reflectivity (LER) model, and

correction for spectral dependence of LER, and the use of GEMS Level-2 (L2) Cloud product. The algorithm is now flexible enough to handle additional wavelengths and more readily employ different sources of a priori profile information without significant changes to the design of the algorithm. The GEMS-O3T algorithm also uses the optimal estimation method (OEM) to make error analysis more accessible and robust.

This paper consists of five Sections: Section 2 describes the GEMS instrument and Level-1B data; Section 3 explains the
differences and advantages of the GEMS algorithm from its predecessor, the TOMS algorithm; Section 4 discusses the results of retrieval characteristics and error analysis; Section 5 presents the validation results of the new TCO product with respect to ground-based Pandora TCO measurements and other satellite TCO measurements from OMPS and TROPOMI; Section 6 discusses the impact of the new algorithm on global TCO observations, and is followed by a conclusion in Section 7.

## 2. Data and Method

### 2.1 The GEMS Mission

GEMS is a UV-visible spectrometer developed for Korea's next-generation geostationary multi-purpose satellite program, which consists of two satellites, Geo-Kompsat-2A (GK-2A) and Geo-Kompsat-2B (GK-2B). They are collocated at 128.2 E over the equator. The GK-2A satellite is equipped with an Advanced Meteorological Imager (AMI) to provide high-resolution images of the Earth's surface and atmosphere for weather forecasting, while GK-2B has two payloads: one with the GEMS
sensor to monitor the atmospheric composition and air quality, and another with Geostationary Ocean Color Imager (GOCI)-II to monitor ocean color.

GEMS is designed to use the same optical path for direct solar radiation and radiance backscattered from the Earth's surface and atmosphere. Using the same optical path for solar irradiance and radiance backscattered from Earth has several benefits. First, it minimizes calibration uncertainty in algorithms using the ratio of radiance to solar irradiance because sensor errors
common to the radiance and irradiance measurements cancel. The measured light passes through the same calibration assembly and scan mirror, telescope, spectrometer, and detectors, which minimizes the possibility of inconsistency between measurements. However, a diffuser is used for solar irradiance measurements, and is located in front of the scan mirror, and introduces a source of calibration error that does not cancel in the radiance to irradiance ratio. The magnitude of this error is difficult to quantify and requires both in-flight calibration measurements and theoretical calculations. We will discuss the
impact of this error on retrieval and how to correct it in Section 4.

GEMS measures Earth radiance in the 300-500 nm wavelength range with a high spectral sampling of 0.2 nm and spectral resolution of 0.6 nm. The spatial resolution of the instrument is 3.5 km × 7 km over Seoul, and the overall field of regard (FOR) is from 5◦ S to 45◦ N latitude and between 75 ◦E to 145 ◦E longitude every hour from 09:00 to 17:00 Korea Standard Time (KST). Solar irradiance is measured over the same wavelength range once per day in the night-time darkness. The incident
light from the telescope is dispersed onto a single two-dimensional charge-coupled device (CCD), which has 1033 spectral

pixels and 2048 pixels in the spatial dimension. A two-axis mirror scans from east to west with a fixed north-south field of view, during 30-minute observation periods which collect measurements across the entire FOR

## 2.2 The GEMS algorithm

A major objective of this study was to obtain total ozone data from geostationary orbit for the first time using the UV
spectrum. The GEMS-O3T algorithm was developed based on the well-researched NASA TOMS algorithm, which is the oldest and most proven method of satellite total ozone retrieval algorithms developed by Dave and Mateer (1967). Since several others have documented earlier versions of the TOMS algorithm over a half-century of development (Bhartia and Haffner, 2012; Bhartia and Wellemeyer 2002; Dave and Mateer, 1967; Haffner et al., 2015; Klenk et al., 1982; Mcpeters et al., 1996), the important goals of using the TOMS algorithm for GEMS is to obtain the most stable and reliable total ozone output.
Because the TOMS algorithm was only applied to total ozone retrievals from a sun-synchronous orbiting satellite, we conducted a series of studies to improve the total ozone data quality with the GEMS-O3T algorithm. The flowchart represents the total ozone retrieval process from the improved GEMS-O3T algorithm in Figure 1. The algorithm consists of two main components: a forward model that calculates the top of atmosphere (TOA) radiance and an inverse model that derives total ozone from the measured radiance.

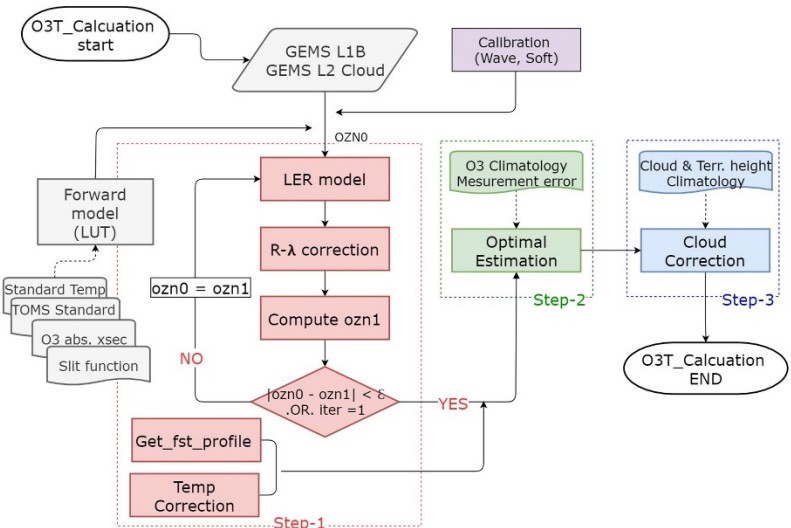


**Figure 1. Flowchart of GEMS-O3T retrieval algorithm, consisting of a forward model for TOA radiance calculation and an inverse model for total ozone derivation. Step 1-3 are highlighted with pink, green, and blue color, respectively.**

### 2.2.1 Forward model

The top-of-atmosphere radiance at the seven wavelengths (312.34, 317.35, 331.06, 340, 354, 360 and 380 nm) is calculated by the VLIDORT radiative transfer model (RTM) (Spurr, 2008). We used a pre-calculated radiance lookup table since performing the VLIDORT calculations online is time-consuming for an operational algorithm. The pre-calculated radiances are obtained at different solar zenith angles, satellite viewing angles, and reflecting surface conditions (land/ocean, clouds, and aerosols) for TOMS standard ozone profiles that vary with latitude band and total ozone amount (Bhartia and Wellemeyer,

2002, Wellemeyer et al., 1997). Due to the limited observational range of GEMS, which covers only low and mid-latitude regions, we employed a reduced set of 11 ozone profiles out of the 21 TOMS standard profiles in our radiance calculations. The surface underlying the atmosphere is assumed to have the Lambert equivalent reflectivity (LER) which treats surfaces, clouds, and aerosols as Lambertian reflectors at terrain pressure (Ahmad, 2004). Our VLIDORT radiance calculations consider polarized Rayleigh scattering and the $O_3$ absorption with temperature-dependent gaseous absorption cross-sections. This study

used the BDM ozone absorption cross-section (Daumont et al., 1992; Brion et al., 1993; Malicet et al., 1995). VLIDORT is also used when calculating the look-up-table (LUTs) for Jacobians, which are needed to perform the retrieval using optimal estimation. We used a single US standard temperature profile to optimize the table size for radiances and Jacobians. Calculated radiances are then adjusted using a zonal mean temperature climatology via a temperature correction in the algorithm. Supplementary sections provide an elaborate account of the radiance lookup tables (LUTs) used in the GEMS-O3T algorithm,

as well as an evaluation of the errors that arise during LUTs interpolation.

### 2.2.2 Inverse Model

An inverse model in the GEMS-O3T algorithm is a mathematical tool that helps to convert the measured radiance into geophysical parameters, such as total ozone and ozone profile. The model proceeds in three steps. Details of the individual

steps are presented below. In Step 1, the reflectivity is derived at 380 nm, then corrected by the method suggested by Dave (1978), followed by the first guess estimate of ozone with 317.35 nm, and finally, residuals and jacobians are calculated. Step 2 is a straightforward implementation of an optimal estimation method to estimate the ozone profiles using inputs derived in Step 1 and a set of radiances (312.34, 317.35, and 331.06 nm) and *a priori* ozone profiles and their error covariance matrix. This process, which is not present in the TOMS-V8 algorithm, is the core of the GEMS-O3T algorithm because it provides the

error amount for retrieved total ozone and the degree of freedom that shows the independent vertical information of the ozone profile. The correction for clouds and terrain height is made in the final process of Step 3.

Step 1 process starts with the computation of reflectivity using the measured BUV radiance at 380 nm based on the simple Lambert Equivalent reflectivity (SLER) model. The initial assumptions are that the spectral dependence of reflectivity ($R$) is zero (i.e., dR/dλ = 0). However, this assumption can no longer be valid in the presence of absorbing aerosol, sea glint, and

clouds (dR/dλ ≠ 0). The algorithm accounts for radiative effects of aerosols and surface reflectance by using the calculated

spectral slope of $dR/d\lambda$ obtained from reflectivity at 340 nm and 380 nm with negligible ozone absorption cross-sections in equation (1).

$$R = R_{380} + \frac{dR}{d\lambda}(\lambda_{317} - \lambda_{380}) \tag{1}$$

This calculation updates the estimated reflectivity ($R$) derived after each iteration proposed by Dave (1978). However, in the presence of high amounts of UV-absorbing aerosols, $dR/d\lambda$ cannot be linear and results in a significant error in the derived reflectivity. This data is flagged during the quality control process. The dependence of the backscattered radiance on ozone is approximately exponential. A quantity called the N-value is defined to reduce the dynamic range of the total ozone dependence (Klenk et al., 1982). The N-value is defined as:

$$N = -100 \log_{10} \frac{I}{F} \tag{2}$$

where $F$ is the extraterrestrial solar irradiance. The amount of total ozone is determined when the measured N-value ($N_m$) at 317.35 nm is equal to the calculated one ($N_c$) with the total ozone amount corresponding to the TOMS standard ozone profile at a given satellite viewing geometry, solar zenith angle, surface reflectivity, and surface pressure (Bhartia and Wellemeyer

2002). The interpolation entails using three to eight ozone profiles, each with a different range of total ozone amounts for latitude. Therefore, the ozone profile shape corresponding to the retrieved total ozone is obtained by this process.

    The OEM approach proposed by Rodgers (2000) is applied in Step 2 to retrieve a coarse ozone profile, estimated from a set of radiances at three wavelengths with different amounts of ozone absorption (312.34, 317.35, and 331.06 nm) using a priori profiles and their error covariance matrices. The use of optimal estimation allows smooth transition between the use of the

different wavelengths, and thereby eliminates the discontinuities associated with TOMS ozone distribution that occurs when the solar angle is large.

$$\hat{x} = x_a + S_aK^T(KS_aK^T + S_e)^{-1}[(N_m - N_1) + K(x_1 - x_a)];$$
$$G = S_aK^T(KS_aK^T + S_e)^{-1};$$
$$A = GK;$$
$$\hat{S} = S_a - S_aK^T(KS_aK^T + S_e)^{-1}KS_a \tag{3}$$

where $\hat{x}$ is the optimized ozone profile consisting of 11 Umkehr layers. The pressure at the bottom of these layers decreases by a factor of 2 starting from the mean sea-level pressure (1013.25 hPa) to 0.99 hPa. The top layer goes from all altitudes above the 0.99 pressure level. $x_l$ is the ozone profile retrieved in Step 1. The $x_a$ is the a-priori ozone profile obtained from ML

climatological profiles (McPeters and Labow, 2012) consisting of 12-month and 18 latitudinal bands with 10-degree intervals from 90° S to 90° N. The $S_a$ is a-priori error covariance matrix ($11 \times 11$ covariance matrix) derived from the ML climatological profile, where correlation is limited to one layer from the diagonal entry. $N_m$ is the measurement radiance vector

consisting of three elements from 312, 317, and 331 nm, and $N_1$ is the N-value calculated from $x_1$. $S_e$ is the measurement error covariance matrix, and is assumed to be a diagonal matrix where the elements are the squares of the assumed measurement errors. We assume the measurement error of 0.12 % according to GEMS SNR corresponding to 320 nm. $K$ is the Jacobian matrix, defined as $dN_j/dx_i$ for each layer $i$ and wavelength $j$, representing partial derivatives of the forward model to the ozone. The gain matrix, $G$, measures how sensitive the retrieved profile is to measurement errors, and the averaging kernel matrix, $A$, provides the sensitivity of the retrieval to a change in true ozone profile and a vertical resolution of the retrieved profile. The covariance matrix $\hat{S}$ measures the degree of error in the retrieved ozone profile. It contains the measurement and smoothing errors propagated through the $G$ and $A$ matrices, respectively. The optimal estimation technique also offers crucial parameters for error analysis, including the degrees of freedom for signal (DFS), retrieved column's estimated error, and column weighting functions (CWFs). The supplement section furnishes a comprehensive elucidation of these variables.

The total ozone obtained from Step 2 assumes that the reflecting surface is at sea level. If the surface is at a different elevation or a cloud is present, we must account for that in the total ozone calculation. After adjusting the CWFs ($w_l$) for the profile used, the final Step 3 total ozone is calculated by subtracting the amount of ozone corresponding to the difference between the topography (or cloud) height and the ground surface from the Step 2 total ozone. Since the ozone column below the cloud pressure ($p_c$) is relatively small, we use a relatively simple method to correct it. If reflectivity ($R$) from 380 nm is less than 0.05 ($R_s$) or snow or ice is present, no cloud is assumed. If $R$ is greater than 0.4 ($R_c$), we assume the entire pixel is covered with clouds. For $R_s < R < R_c$, the pixel was assumed to be partial cloud cover, and the cloud fraction was determined by equation (4) as follows:

$$f_c = (R - R_s)/(R_c - R_s). \tag{4}$$

We assume that this fraction ($f_c$) is approximately the fraction of the measured radiance signal reflected by clouds within the instrument field-of-view. We first estimate the ozone between the cloud and terrain pressure in each layer $l$, and then set the ($w_l$) in layers below $p_c$ to zero in equation (5)

$$x_l^* = \hat{x}_l(1 - f_c) + x_{a,l}f_c,$$
$$w_l^c = w_l(1 - f_c), \tag{5}$$

where $\hat{x}_l$ and $x_{a,l}$ are the ozone amount in layer $l$ of the profile obtained from step 2 and from a priori profile respectively. Then, the correction to the total column is obtained by

$$\delta\Omega = \sum_{l=0}^{l_{pc}} x_l^* (1 - w_l^c) - \hat{x}_l (1 - w_l^c). \tag{6}$$

## 2.3 Correlative Satellite Measurements

OMPS was launched in October 2011 on the Suomi National Polar-orbiting Partnership (SNPP) satellite and includes both nadir- and limb-viewing modules. OMPS NM total ozone data (OMPS NMTO3) were used in this study. The OMPS NM is a hyperspectral imaging push-broom sensor with a 110° cross-track field of view (FOV), and 35 cross-track positions. OMPS NM has a $50 \times 50$ km$^2$ spatial resolution at the nadir and measures solar backscattered ultraviolet radiation in the spectral range from 300 to 380 nm. The OMPS total ozone algorithm is based on the NASA version 8 total ozone algorithm (Bhartia and

Wellemeyer, 2002). In our study operational OMPS-NM Level 2 (L2) version 2.1 were used. As validated in McPeters et al. (2019), the maturity of this product is high with biases of less than 0.2 % when compared to ground-based measurements in the Northern Hemisphere.

TROPOMI was launched in October 2017 on the Sentinel-5 Precursor (S5P) satellite. TROPOMI aboard S5P is a nadir viewing spectrometer that provides measurements in the ultraviolet, visible, near-infrared, and shortwave infrared spectral

bands. TROPOMI has a swath width of 2600 km (roughly 104° wide) with a ground pixel resolution of 3.5 km $\times$ 5.5 km (Veefkind et al., 2012). S5P/TROPOMI offline (OFFL) total ozone column products were used in this study which are obtained using the GODFIT version 4 retrieval (Lerot et al., 2021, Spurr et al., 2022). The algorithm directly compares with simulated radiances through nonlinear least-squares inversion using the sun-normalized measured radiance from 325 to 335 nm. The calculated radiances and Jacobians are obtained with the RTM LIDORT (Spurr et al., 2008). A validation for S5P/TROPOMI

OFFL TOC with global ground-based measurements from April to November 2018 was found to be well within acceptable limits, with mean biases (MB) ranging from 0% to 1.5% and standard deviations between 2.5% and 4.5% for monthly mean co-locations (Garane et al., 2019).

## 2.4 Correlative Ground-based Measurements

The Pandora TCO retrieval algorithm utilizes a modified version of the Differential Optical Absorption Spectroscopy (DOAS)

technique to determine the concentration of atmospheric constituents. In the case of TCO, the DOAS method compares direct solar spectra measured by the Pandora spectrometer to an independent extraterrestrial reference spectrum, which represents the expected solar spectrum in the absence of atmospheric absorption. Through spectral analysis of the measured and reference spectra within the 305 to 328.6 nm wavelength range, the Pandora algorithm retrieves TCO values using a spectral fitting approach, wherein fitting parameters are optimized to minimize the difference between the measured and modeled spectra.

Additionally, the Pandora algorithm accounts for the effects of Rayleigh scattering and atmospheric absorption species such as $NO_2$ and $O_4$. Technical details about the retrieval algorithm and configuration settings are available in the software manual (Cede et al., 2021). The TCO used in this study was processed and retrieved by using Blick software Suite (version 1.7).

# 3. Results and Discussion

## 3.1 GEMS hourly total ozone distribution

The GEMS sensor onboard a geostationary satellite has the advantage of measuring ozone over current Low Earth Orbiting (LEO) satellite sensors because it provides hourly observations throughout the data, which helps to improve our understanding of ozone. GEMS V2.0 total ozone products are used in our analysis. To assess the performance of the GEMS total ozone algorithm, the hourly GEMS total ozone on 29 March 2021 is shown in Figure 2. This figure shows the eight hourly GEMS total ozone measurements from 09 Korea Standard Time (KST) to 16 KST. Because GEMS measures backscattered UV radiation when the solar zenith angle is not large, daytime hourly observation time and area vary depending on the season (NIER, 2020). Figure 2(a) shows the hemisphere east (HE) mode at 9:00 KST, Figure 2(b) shows the hemisphere Korea (HK) mode at 10:00-11:00 KST, Figure 3(c)-(d) shows the full central (FC) mode at 11:00-12:00 KST, and Figure 2(e)-(h) shows the full west (FW) mode at 13:00-16:00 KST. The GEMS observation usually takes 30 minutes, running from 15-minutes before the hour to 15-minute after the hour.

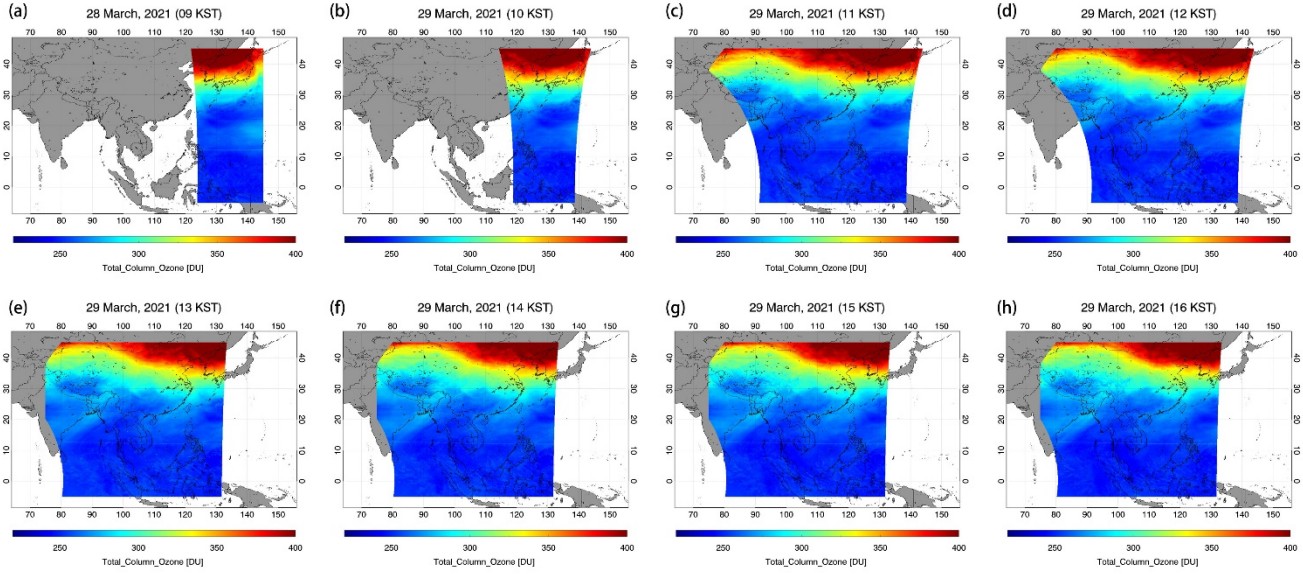

**Figure 2. Hourly GEMS total ozone distribution on 29 March 2021.**

The total ozone distribution ranging from 250 to 400 DU shows a typical distribution in March: High values at high latitudes, followed by a sharp decrease in the middle latitudes and gradually decreasing toward the equator. Because the scale bar is so large, changes in hourly values are not clearly seen. The GEMS hourly ozone monitoring system provides continuous updates on stratospheric ozone and its associated atmospheric changes. It also provides essential information to models that help us predict the future development in the ozone state.

Figure 3 compares the hourly GEMS TCO with Pandora TCO observed over eight ground sites and satellite TCO for three consecutive days from 29 to 31 March 2021. The hourly total ozone distribution in Figure 3 showed significant diurnal ozone changes of up to 40 DU. This indicates that the ozone undergoes significant diurnal change primarily due to changes in stratospheric ozone, and is evidence of why hourly ozone monitoring is important to track dynamic ozone changes. Pandora TCO varies considerably over time, and the diurnal variation of GEMS is in good agreement with that of Pandora. The GEMS data for diurnal ozone change offers advantages over TROPOMI (blue) and OMPS (green) ozone data, which are observed once per day.

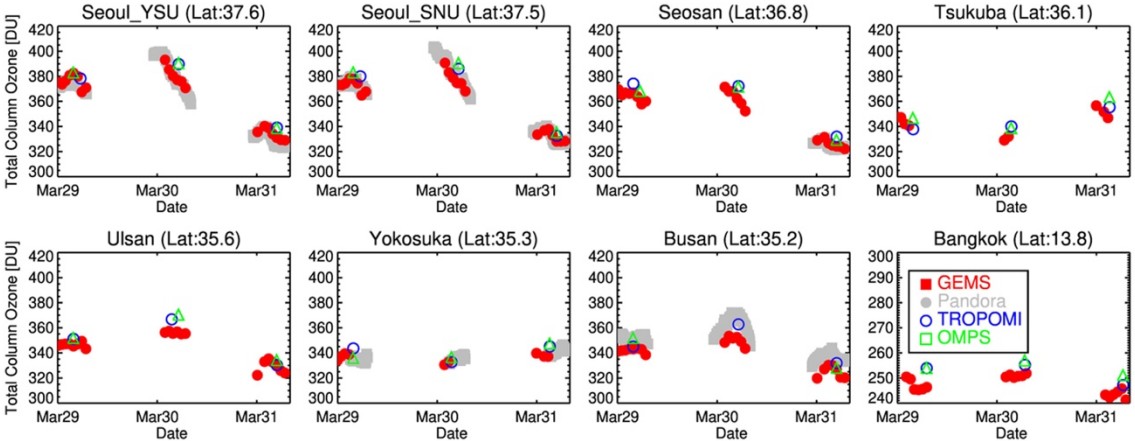

**Figure 3. Comparison of GEMS TCO with Pandora, TROPOMI, and OMPS TCO from 29 to 31 March 2021 over eight Pandora sites. The TCO measurements are represented using red filled squares for GEMS, gray filled circles for Pandora, blue empty circles for TROPOMI, and green empty squares for OMPS.**

## 3.2 Validation of GEMS Total Ozone Measurements with Pandora

Satellite measurements are subject to instrument measurement errors and retrieval errors from ill-posed problems. Therefore, validation is essential for scrutinizing satellite retrieval accuracy and providing confidence in the final results. The GEMS total ozone data was validated by comparing it with ground-based Pandora and other satellite measurements from OMPS and TROPOMI. For accurate validation, we used GEMS TCO between August and December 2020, a stable initial operation period with accurate Image Navigation and Registration (INR) information. The available Pandora observations during this period were over Busan, Ulsan, Seoul, and Yokosuka. Table 1 presents detailed Pandora site information. Since GEMS switches to Full West mode at 13 KST in November and December, there is no GEMS measurement at Yokosuka in Japan from this time, so we used only data before this time for validation. GEMS takes 30 minutes to complete an observation over the FOR, whereas Pandora collects each measurement in 2 minutes several times per day. For temporal coincidence, we used the average of Pandora observation data before and after 15 minutes of the local GEMS observation. For spatial coincidence,

we used the closest GEMS data to the Pandora site. To exclude Pandora data contaminated by clouds and aerosols, we used data with the normalized root-mean-square (RMS) of weighted spectral fitting residuals less than 0.05 % and the estimated error in TCO less than 2 DU as suggested by (Tzortziou et al., 2012). We take GEMS data with a solar zenith angle of less than 75° to avoid GEMS errors that may occur due to the high solar zenith angle of GEMS data.

285 **Table 1. Pandora observation sites over the GEMS comparison domain.**

| Site name | Longitude | Latitude | Period |
|---|---|---|---|
| Busan, Korea | 129.1 | 35.2 | 2020/08/01 ~ 2020/12/28 |
| Ulsan, Korea | 129.2 | 35.6 | 2020/08/01 ~ 2020/11/04 |
| Seoul, Korea | 127.0 | 37.5 | 2020/08/18 ~ 2020/12/31 |
| Yokosuka, Japan | 139.7 | 35.3 | 2020/10/29 ~ 2020/12/31 |

Figure 4 represents the comparison of GEMS, TROPOMI, and OMPS with Pandora TCO at the sites. The comparison between GEMS and Pandora was performed at 0345 and 0445 UTC, which correspond to the overpass time of TROPOMI/OMPS, in order to exclude potential errors that vary throughout the day and were not taken into account in this 290 comparison. Figure 4(a) shows a high correlation of 0.97 or more with GEMS and Pandora TCO at Seoul, Busan but a low correlation of 0.90 at Ulsan, which is significantly smaller than at other sites. RMSE showed satisfactory small values, with the lowest RMSE of 1.3 DU. As mentioned earlier, because GEMS operates in Full West mode starting at 13 KST during November and December, and there are no GEMS measurements at the overpass time of TROPOMI/OMPS in Yokosuka, Japan. Mean biases (MBs) ranging from -1.36 to 0.76% were observed at all sites, with the highest positive MBs occurring in 295 August. The MBs showed a distinctly high value in August (summer) with 3.30% in Busan and 2.87 % in Seoul, then decreased to 0.54% and -1.36% in December (winter), respectively. Overall, it is noteworthy that the mean biases (MBs) of GEMS-Pandora decrease significantly over time, decreasing from August to October before slightly increasing in December

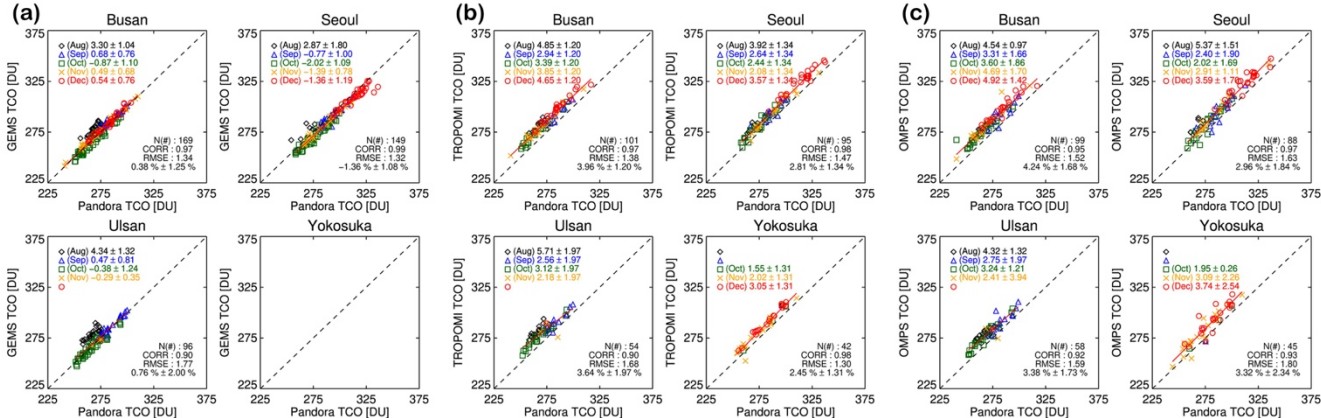

**Figure 4.** Scatter plots of Pandora TCO with (a) GEMS TCO, (b) TROPOMI TCO, and (c) OMPS TCO at Busan, Seoul, Ulsan, and Yokosuka. A linear fit representing a 1:1 ratio is shown in black dotted lines. The legends of N, CORR, RMSE and the number in percentage represent the number of data points, correlation coefficient, RMSE, and mean bias with standard deviation, respectively. The number on the bottom right is the percentage bias for each month. The comparison between GEMS and Pandora was conducted at the overpass time of TROPOMI/OMPS to eliminate potential errors that vary over the course of the day and were not included in this comparison. As mentioned earlier, because GEMS operates in Full West mode starting at 13 KST during November and December, and there are no GEMS measurements at the overpass time of TROPOMI/OMPS in Yokosuka, Japan.

Figures 4(b) and 4(c) show the comparison of Pandora with other satellite data, TROPOMI and OMPS. Their correlation is similar to that of GEMS and Pandora. The correlation of Pandora with OMPS and TROPOMI is the lowest in Ulsan, 0.92 and 0.90, respectively. There seems to be an issue with the Pandora measurements at Ulsan. The RMSE between Pandora and both satellites were less than 2 DU, as in GEMS.

**Table 2.** The statistical metrics, including correlation coefficient (R), root mean square error (RMSE), mean bias (MB), and mean standard deviation errors (MSE) comparing GEMS, TROPOMI, and OMPS with Pandora TCO at Busan, Seoul, Ulsan, and Yokosuka sites

| | N | R | RMSE [DU] | MB [%] | MSE [%] |
|---|---|---|---|---|---|
| GEMS | | | | | |
| Busan | 169 | 0.97 | 1.34 | 0.38 | 1.25 |
| Seoul | 149 | 0.99 | 1.32 | -1.36 | 1.08 |
| Ulsan | 96 | 0.9 | 1.77 | 0.76 | 2 |
| TROPOMI | | | | | |
| Busan | 101 | 0.97 | 1.38 | 3.96 | 1.2 |
| Seoul | 95 | 0.98 | 1.47 | 2.81 | 1.34 |
| Ulsan | 54 | 0.9 | 1.68 | 3.64 | 1.97 |
| Yokosuka | 42 | 0.98 | 1.3 | 2.45 | 1.31 |

| OMPS | | | | | |
|---|---|---|---|---|---|
| | N | R | RMSE [DU] | MB [%] | MSE [%] |
| Busan | 99 | 0.95 | 1.34 | 4.24 | 1.68 |
| Seoul | 88 | 0.97 | 1.63 | 2.96 | 1.84 |
| Ulsan | 58 | 0.92 | 1.59 | 3.38 | 1.73 |
| Yokosuka | 45 | 0.93 | 1.8 | 3.32 | 2.34 |

Although no monthly trend was observed as distinct as GEMS, the MB in TROPOMI and OMPS increased in August
(summer) and December (winter). The reason for this is that, as Herman et al., (2015) showed, the satellite retrieval methods
perform temperature correction for the temperature-sensitive ozone absorption coefficient, whereas Pandora uses a fixed-
temperature ozone absorption coefficient. Therefore, comparisons of satellite and Pandora data may show seasonal dependence.
However, the seasonal variability shown in the comparison of GEMS and Pandora differs from those between TROPOMI
(OMPS) and Pandora in magnitude and seasonal dependence.

Figure 5 is a time series showing the percentage difference between three satellite observations (GEMS, TROPOMI, and
OMPS) and Pandora. The overall mean bias for TROPOMI-Pandora and OMPS-Pandora is within 3.8 % for all stations, which
is consistent with the previous studies (Herman et al., 2015). As for the mean standard deviation, TROPOMI has lower
variability in comparison to OMPS. This could be due to the lower spatial resolution of OMPS at 50 km by 50 km compared
to TROPOMI at 5.5km by 3.5 km. In the case of Ulsan, both comparisons of TROPOMI and OMPS with Pandora showed a
low correlation (~0.90) and a high standard deviation (~1.8 %) compared to other stations. These comparison results suggest
the Pandora measurement at Ulsan suffers from problems in the accuracy of total ozone measurement which may be due to
some form of instrument error. Therefore, we have excluded the Pandora measurements at Ulsan from a reference dataset for
further GEMS validation at this time. The mean bias of GEMS with Pandora is 0.11% with a standard deviation of 2.17% for
all stations, and the comparison with Pandora observations in the Yokosuka area shows a slightly lower bias of -2.96% than
the comparison results in other areas. However, it is not appropriate to draw any conclusions by comparing GEMS with
Yokosuka using only 24 data points over 5 months.

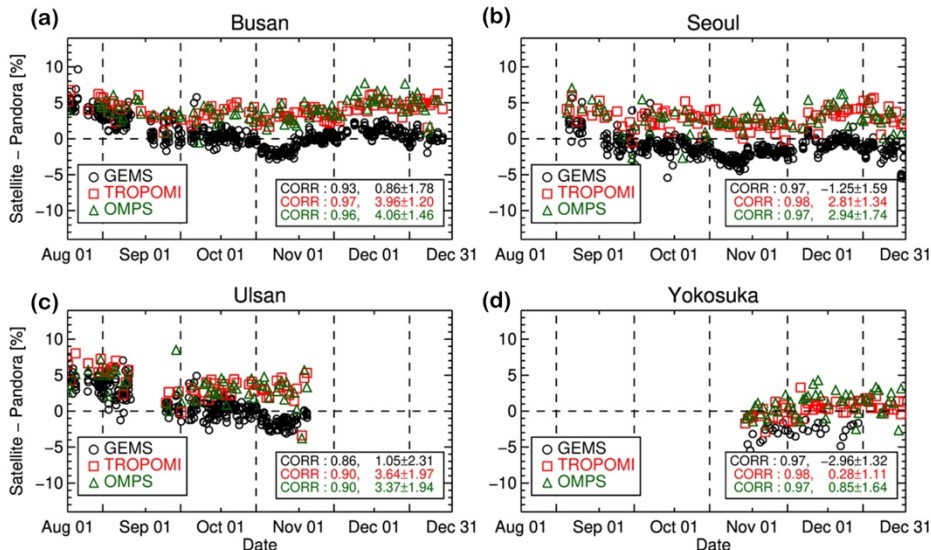

**Figure 5. Time series of the daily percentage difference between Pandora and three satellite observations (with GEMS in black, TROPOMI in red and OMPS in green) at Busan, Seoul, Ulsan, and Yokosuka from August to December 2020.**

In Busan and Seoul, the mean bias (MBs) is highest in August, with 3.5% and 2.1%, respectively. The MBs then decreases from September to October before slightly increasing again in December. This seasonal pattern, although slightly overestimated in August, is similar to the MBs of TROPOMI and OMPS (Figure 4b, Figure 4c). GEMS overestimates by ~0.85% in Busan and underestimates by -1.25% in Seoul compared to Pandora over the entire analysis period. The significant bias observed in August does not appear to have a substantial impact on the average bias due to the reduced sample size

resulting from cloud filtering. The comparison of the temporal distribution of GEMS and Pandora in Figure 6 shows that GEMS can observe ozone daily variations that are nearly identical to Pandora within a similar range of errors although GEMS has a bias of approximately 1% compared to Pandora.

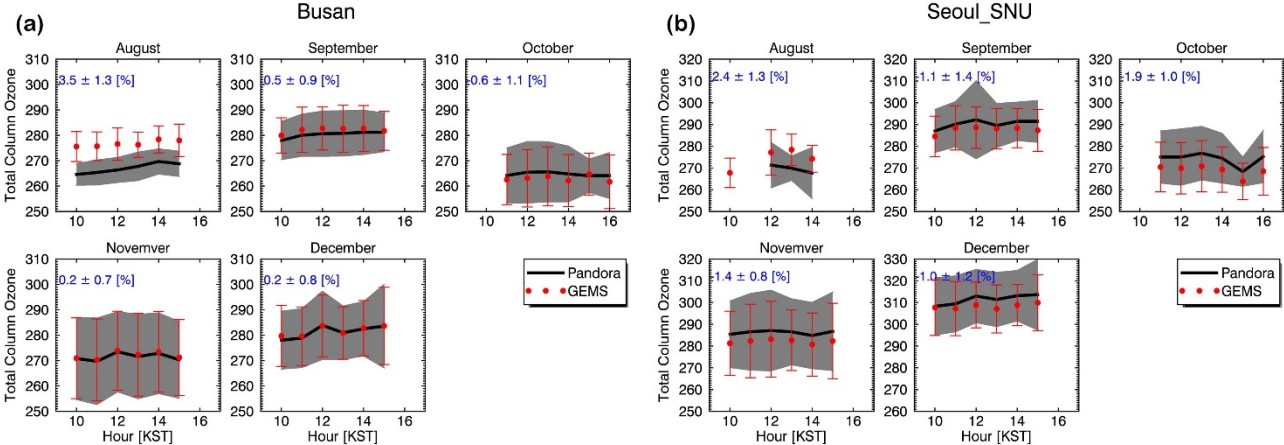

**Figure 6. Time variation of monthly mean values of GEMS (red filled circles) and Pandora (a black solid line) in Busan and Seoul covering the period from August to December. The standard deviation of Pandora TCO is represented by the gray shading, while the standard deviation of GEMS TCO is indicated by the bars.**

## 3.3 Validation of GEMS total ozone with other satellites

Figure 7 shows the spatial distribution of TCO from GEMS, TROPOMI, and OMPS in the GEMS domain on 30 November 2020. This figure shows that the spatial distribution of TCO observed from the three satellites is in good agreement. It shows a typical ozone distribution pattern that increases from low to high latitudes.

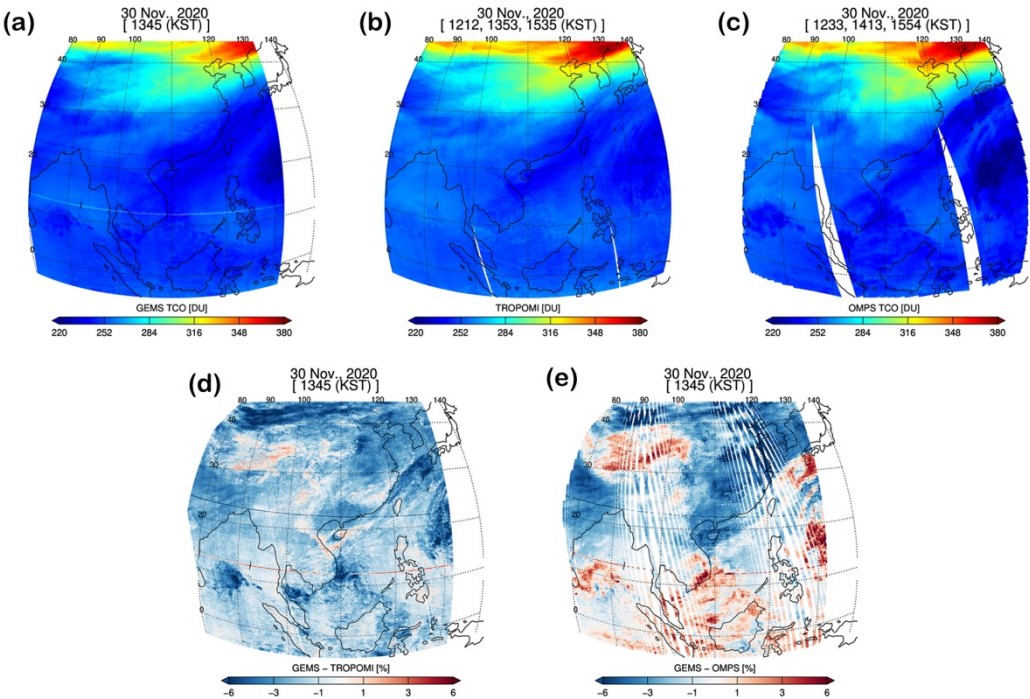

**Figure 7. Maps of total column ozone from (a) GEMS, (b) TROPOMI, (c) OMPS, (d) percentage difference between GEMS and TROPOMI, (e) percentage difference between GEMS and Pandora on 30 November 2020.**

The distribution of wave patterns at high latitudes appears to be caused by atmospheric dynamics associated with meteorological phenomena. The horizontal striping in GEMS found around 10° and 20° latitudes is an error caused by the bad pixels of the GEMS detector. These bad pixels are expected to be removed properly in the future by using an improved bad pixel mask variable in the GEMS level 1C data. Figure 7(d) and Figure 7(e) display the bias maps of GEMS with respect to TROPOMI and OMPS, respectively. In Figure, GEMS TCO consistently shows a -3% bias compared to the TCO from both

satellites. However, Figure 7(e) reveals a distinct positive bias of 2-3% that is not evident in Figure 7(d). This positive bias is observed in the high reflectivity region associated with clouds, as indicated in Figure 8. It is particularly pronounced in areas where OMPS measures significantly lower ozone, as shown in Figure 7(c). The strong anti-correlation between total ozone and clouds can be attributed to the difference in cloud height estimation methods used by the OMPS algorithm compared to GEMS and TROPOMI. OMPS derives cloud height from cloud climatology (Joiner and Vasilkov, 2006) while GEMS and

TROPOMI retrieve cloud information from real-time calculated cloud L2 products. The GEMS cloud retrieval algorithm employs the Differential Optical Absorption Spectroscopy (DOAS) method with the O2-O2 absorption band to retrieve effective cloud fraction, cloud centroid pressure, and cloud radiance fraction (NIER, 2020a). On the other hand, TROPOMI utilizes two algorithms for cloud retrieval: OCRA (Optical Cloud Recognition Algorithm) and ROCINN (Retrieval of Cloud Information using Neural Networks). OCRA estimates cloud fraction by analyzing TROPOMI measurements in the ultraviolet

and visible spectral regions, while ROCINN uses TROPOMI measurements within and around the oxygen A-band in the near
     infrared to retrieve cloud top height (pressure) and optical thickness (albedo). For more detailed information on these cloud
     algorithms, refer to NIER (2020a) and Loyola (2018). This difference in cloud height affects TCO retrieval. Only ozone present
     above the cloud can be retrieved from the satellite's UV radiance over the cloudy scene, resulting in column ozone from the
     cloud height to the top of the atmosphere. The final TCO is calculated by adding the climatological ozone corresponding to

the lower part of the cloud height. The OMPS climatology cloud height, as depicted in Figure 8(c), is remarkably lower than
     the cloud height retrieved by GEMS and TROPOMI in regions where actual clouds are present. Consequently, applying a
     lower cloud height leads to a reduced amount of ozone added below the cloud, resulting in a smaller OMPS TCO. On the other
     hand, a substantial decrease of approximately -5% in the bias of GEMS for TROPOMI is evident in Figure 7(d), specifically
     in regions characterized by high cloud fraction and altitude (Figure 8). TROPOMI consistently indicates a cloud height of

around 300 hPa, while GEMS retrieves a cloud altitude of approximately 500 hPa, revealing a significant disparity in cloud
     height estimation. Although some differences in cloud altitudes are expected due to the use of different algorithms, the
     observed disparity in cloud height between the two datasets is considerable. Therefore, further research is needed to investigate
     the impact of this significant difference in cloud height on the bias of GEMS for TROPOMI.

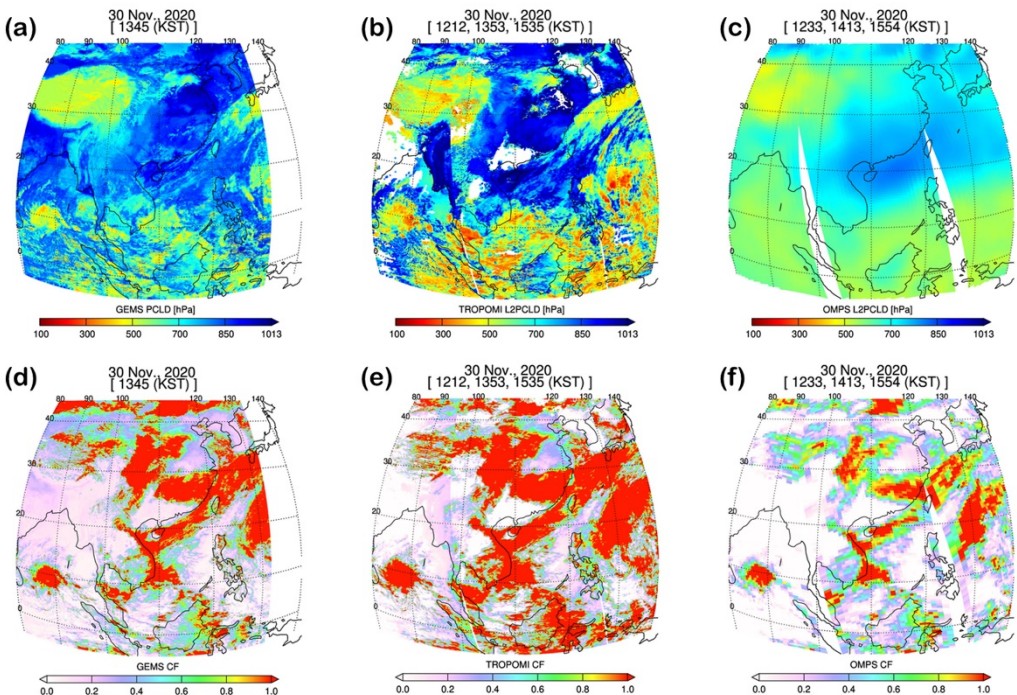

**Figure 8 The spatial distribution of cloud pressure and cloud fraction obtained from GEMS, TROPOMI, and OMPS satellite
     observations on 30 November 2020. Panels (a), (b), and (c) display the maps of cloud pressure derived from GEMS, TROPOMI, and
     OMPS, respectively. Similarly, panels (d), (e), and (f) show the maps of cloud fraction obtained from GEMS, TROPOMI, and OMPS,
     respectively.**

The histogram analysis was performed to compare the data sets with different spatial and temporal resolutions over the
GEMS domain from August to December 2020 (Figure 9). The histogram of all satellite data is similar to the normal
distribution showing good agreement with each other. Especially, the distribution shape of GEMS with an average of 267.3
DU and TROPOMI with an average of 272.6 DU are very similar. However, the average of OMPS is smaller than the two-
satellite data, and the peak is also tilted to a lower side than the average. This appears to be due to low ozone in cloudy pixels,
as mentioned earlier.


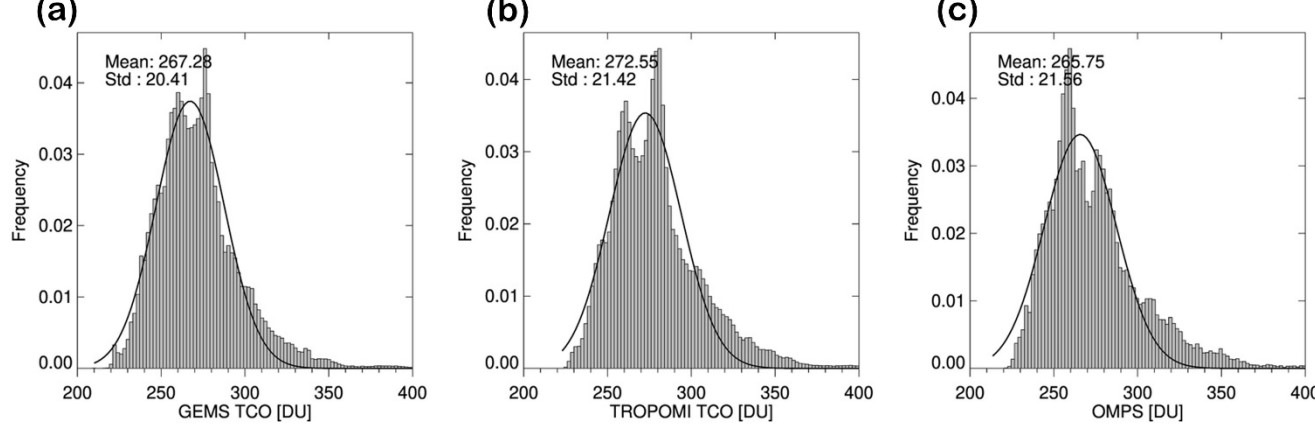

**Figure 9. Histogram distribution of TCO from (a) GEMS, (b) TROPOMI, and (c) OMPS from August 01 to December 31, 2020, with their corresponding Gaussian fitting lines (black).**

Since TROPOMI and OMPS have different observation times and fields of view relative to GEMS, it is necessary to match
the spatial and temporal correspondence of the two data for quantitative comparison. For temporal consistency, the observation
time difference between the polar orbit satellite and GEMS is less than 30 minutes. For spatial consistency, we selected the
closest points within 10 km of the observation point of the two satellites. In addition, to use good quality data for comparison,
we used only data satisfying the quality control conditions presented in Table 3.


**Table 3. Summary of validation data and methods**

|  | **TROPOMI** | **OMPS** |
|---|---|---|
| Validation product | GEMS total column ozone | |
| Validation region | East Asia (75°E - 140°E, 5°S - 45°N) | |
| Validation period | 2020.08.01 ~ 2020.12.31 | |

| Quality control | TROPOMI data used in this study must meet the following criteria:<br>• $0 < TCO < 1008.52$<br>• $180K < To3 < 260\ K$<br>• Ring scale factor $< 0.15$<br>• $-0.5 <$ effective albedo $< 1.5$<br>• $CF < 0.2$ | OMPS data used: algorithm flag = 0 or 1, cloud fraction $< 0.2$<br>Excluded data: cross-track positions between 1 and 35. |
|---|---|---|
| Quality control (GEMS product) | 1. cloud filtering ($CF < 0.2$ from GEMS L2 CLOUD product)<br>2. Use GEMS products correspond to final algorithm flag equal to 0 or 1.<br>3. Exclude GEMS products correspond to GEMS L1C bad pixel mask equal to 1. | |
| Co-location method | Distance difference within 10 km.<br>Time difference $< 30$ min | Distance difference within 25 km.<br>Time difference $< 30$ min |

Figure 10 shows the quantitative comparison of GEMS TCO data with TROPOMI and OMPS TCO data for five months. It shows a high correlation coefficient greater than 0.98 and a low RMSE of less than 1.8 DU over clear sky conditions. Compared to TROPOMI and OMPS, GEMS shows underestimation with a negative bias of -2.38% (6.5 DU) and a standard deviation of 1.33%, and a negative bias of -2.17% (6 DU) and a standard deviation of 1.57%, respectively. It shows that the GEMS TCO agrees very well with the TROPOMI and OMPS TCO. However, in the red circle in Figure 10(a), a distinctly high value for the GEMS TCO is observed compared to the TROPOMI TCO. The reason for this is that we did not remove the amount of $SO_2$ ejected by the volcanic eruption of Nishinoshima (27.247° N, 140.874° E) in Japan between August 1 and 5 from the GEMS TCO, which resulted in a high GEMS TCO. There will be a further discussion about this in Figure 11. Figure 10(b) shows the correlation between GEMS and OMPS. The abnormal deviation shown in Figure 10(a) was not observed. Probably, the $SO_2$ influence was not removed because OMPS and GEMS use a similar algorithm.

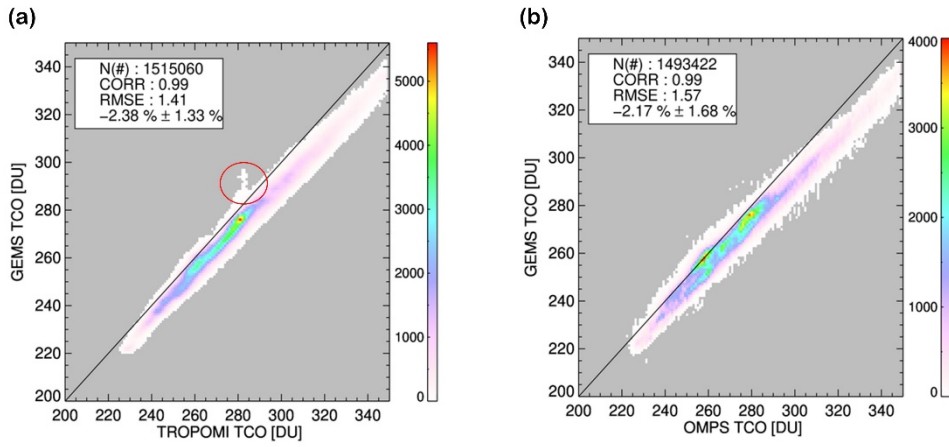

**Figure 10. The comparison of GEMS TCO with (a) TROPOMI and (b) OMPS TCO from August 1 to December 31, 2020.**

Figure 11 shows the distribution of satellite TCO and $SO_2$ on 4 August 2020, the day after the volcanic eruption of Nishinoshima in Japan. GEMS and OMPS show high TCO in regions with high $SO_2$ over 6 DU, but no distinctly high values from the TROPOM TCO are observed. At a wavelength of 317.5 nm, which TOMS-based GEMS and OMPS algorithms use for ozone measurement, $SO_2$ also has a strong absorption line. Therefore, if the $SO_2$ effect was not properly removed, TCO will be overestimated (Fisher et al., 2019; Krueger et al., 2008). However, since the TROPOMI direct-fitting algorithm derives the TCO using a 325-335 fitting window with a weak $SO_2$ absorption band, the $SO_2$ interference is negligible (Spurr et al., 2021).

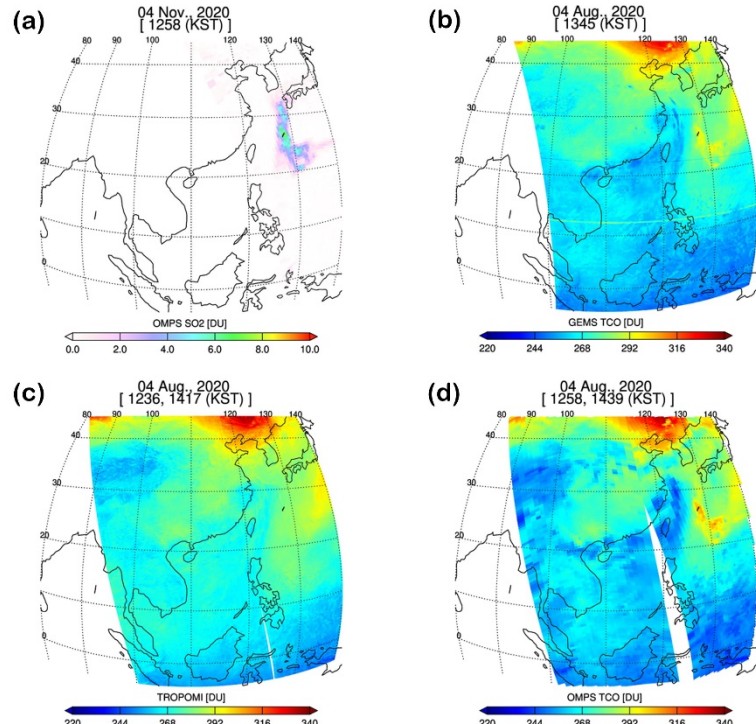

**Figure 11. The map of (a) OMPS SO2, (b) GEMS TCO, (c) TROPOMI TCO, and (d) OMPS TCO in the case of the volcanic eruption of Nishinoshima on 4 August 2020.**

Figure 12 shows the MBs between GEMS and TROPOMI, as well as between GEMS and OMPS as a function of latitude for each month. GEMS-TROPOMI and GEMS-OMPS exhibit mean biases (MBs) of less than 1% at low latitudes, but at mid-latitudes, both MBs become negative and exhibit an increasingly negative dependence on latitude. Moreover, the dependency increases from August to December. The most significant change occurs at 40°N, where the mean bias changes from approximately -1% in August to -4% in December.

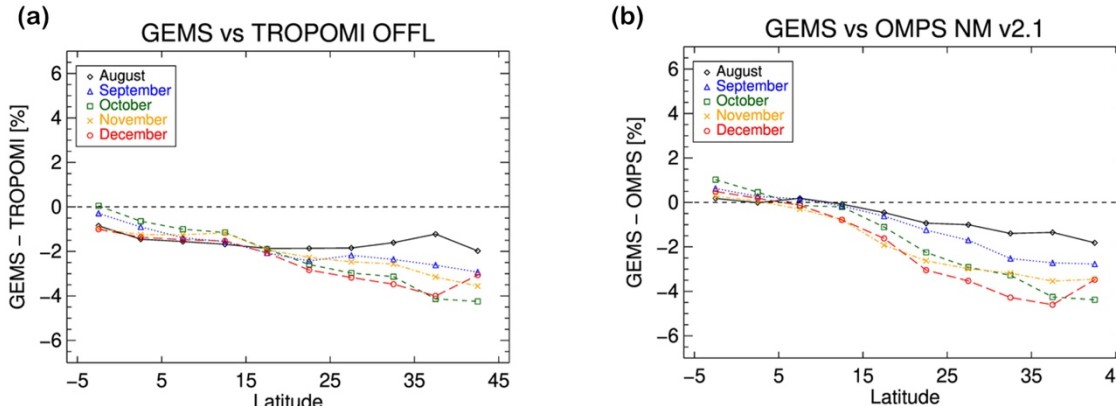

**Figure 12. Mean Bias in TCO between GEMS and TROPOMI (left) and GEMS and OMPS as a function of Latitude and months from August 2020 to December 2020. GEMS retrievals with the algorithm flag equal to 0, or 1 and SZA and VZA <70°. Symbols in the figure denote months as follows: black diamonds for August, blue triangles for September, green squares for October, yellow crosses for November, and red circles for December.**

Kang et al. (2022) noticed a problem in GEMS Level 1C irradiance because the Bidirectional Transmittance Distribution Function (BTDF) of the GEMS diffuser changes depending on the sun illumination angle. They compared the daily GEMS irradiance to the solar reference spectrum, which was obtained from the convolution of the KNMI spectrum (Dobber et al., 2008) with the GEMS spectral response functions (SRF) (Kang et al., 2020). The GEMS irradiance was 20 % smaller than that of the reference spectrum and showed distinct spatial and seasonal variability. An empirical correction was applied to the BTDF to correct the GEMS irradiance by using the azimuthal angle and temporal variation of the GEMS instrument (Kang et al., 2022). We conducted an analysis on the GEMS TCO data calculated using the corrected GEMS irradiance data, following the same analysis method as shown in Figure 12. Figure 13 shows a significant reduction in the MBs of TROPOMI and OMPS, which were 1% and 0% respectively at low latitudes, to -3% and -2%. The apparent decrease seen in mid-latitudes in Figure 12 was also significantly reduced. Although the overall negative bias compared to TROPOMI and OMPS remained consistent, the distinct negative trend seen in Figure 12 for different latitudes and seasons was improved. However, for December in the latitude range of 30°N to 45°N, there is a sudden positive bias trend increasing to -0.5%, which is not found for different months. Therefore, to improve the accuracy of the GEMS ozone algorithm, more research is needed not only on bias correction but also on the performance of GEMS irradiance measurements.

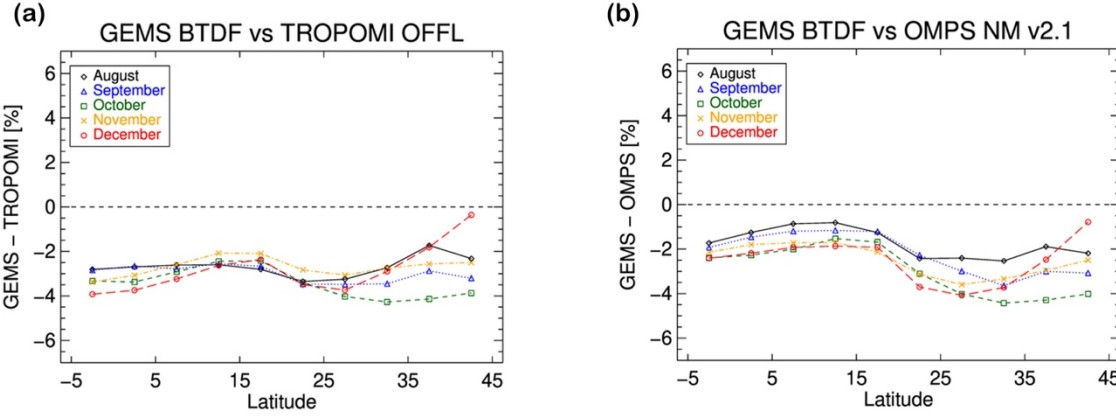

**Figure 13. The Mean Bias (MB) in TCO between GEMS with BTDF correction and TROPOMI (on the left), and between GEMS with BTDF correction and OMPS (on the right), as a function of latitude and month from August 2020 to December 2020. Symbols in the figure denote months as follows: black diamonds for August, blue triangles for September, green squares for October, yellow crosses for November, and red circles for December.**

## 4. Conclusion

The launch of the first geostationary environmental satellite, GEMS, has marked an important milestone in providing hourly monitoring of stratospheric ozone and air pollution, which significantly impact humans and ecosystems. This paper provides the atmospheric science community with the world's first assessment of GEMS total ozone retrieval performance and diurnal ozone variation. The algorithm used for GEMS is a more advanced version of its predecessor, the TOMS-V8 algorithm. In addition to calculating total ozone, it has the advantage of providing ozone profile and retrieval error information.

To assess the performance of the GEMS algorithm, the hourly GEMS TCO was compared with the ground-based TCO measurements from Pandora that vary considerably through the day. The diurnal variation of GEMS total ozone captures this variability and shows good agreement with that of Pandora. This indicates that the ozone undergoes significant diurnal change, primarily due to changes in stratospheric ozone, and is evidence of why hourly ozone monitoring is important to track dynamic ozone changes. For further validation of GEMS TCO, we performed cross-comparisons between GEMS, Pandora TCO and other satellite sensors, OMPS and TROPOMI. GEMS shows a high correlation of 0.97 and low RMSE compared to Pandora TCO at Busan and Seoul, and exhibits daily variations in ozone that are highly consistent with Pandora measurements, with a bias of approximately 1%, despite exhibiting seasonal dependence in the mean bias of GEMS-Pandora. The comparison of GEMS TCO data with TROPOMI and OMPS TCO data shows a high correlation of 0.99 and low RMSE, but a negative bias of -2.38% and -2.17%, respectively, with standard deviations of 1.33% and 1.57%. The influence of $SO_2$ from volcanic eruptions is not properly removed in some regions, leading to GEMS overestimating TCO in those areas, similar to OMPS. The mean biases of GEMS TCO data with TROPOMI and OMPS TCO are less than 1% at low latitudes but become negative at mid-latitudes with an increasingly negative dependence on latitude. Furthermore, this dependence becomes more prominent

from summer to winter. GEMS solar irradiance is 20 % lower than the Dobber et al., (2008) reference spectrum, and shows distinct spatial and seasonal variability. An empirical correction applied to the GEMS irradiance data improved the dependence of mean bias on season and latitude, but a consistent bias still remains, and a marginal positive trend was observed in December. Improvements in GEMS sensor characterization should improve the quality of GEMS total ozone retrieval. Nevertheless, the

results presented in this work that have been achieved thus far are a meaningful scientific advance by providing the first validated, hourly UV ozone retrievals from a satellite in geostationary orbit. This experience can be used to advance research with future geostationary environmental satellite missions, including TEMPO and Sentinel-4 which are planned to launch in 2023.

**Code availability.**

All input data including GEMS measurement and validation measurements related to this paper are available from the corresponding author on reasonable request (jaekim@pusan.ac.kr).

**Data availability.**

1) GEMS data are available through the GEMS Users Data Hub (https://nesc.nier.go.kr/product/), Accessed: [last access: 5 February 2023]. The GEMS V2.0 data used in our study. At present, the NIER website (https://nesc.nier.go.kr/product/).
only provides GEMS V2.0 data from November 2021 onwards. However, data production for the period prior to that is expected to be reproduced and made available soon. In the meantime, if you request GEMS V2.0 data for the period August 2020 to December 2020, we can provide it to you personally.

2) Copernicus Sentinel-5P (processed by ESA), 2020, TROPOMI Level 2 Ozone Total Column products. Version 02. European Space Agency. https://doi.org/10.5270/S5P-ft13p57.

3) Richard McPeters (2017), OMPS-NPP NMTO3 L2 V2.1, Greenbelt, MD, USA, Goddard Earth Sciences Data and Information Services Center (GES DISC), Accessed [last access: 5 February 2023], doi: 10.5067/0WF4HAAZ0VHK.

4) Pandora data are available through the website http://data.pandonia-global-network.org/, Accessed [last access: 5 February 2023].

**Author contributions**

Jae Hwan Kim designed the research and managed this paper. Kanghyun Baek conducted the algorithm development and validation, and writing of the original draft. Juseon Bak contributed to the analysis of errors. David Haffner worked on the development of the GEMS algorithm. Mina Kang substantially contributed to the analysis of GEMS level-1 data. Hyunkee

Hong curated a variety of data sources. All the co-authors provided comments and contributed to editing the manuscript and figures.


**Competing interests.**

The contact author has declared that none of the authors has any competing interests.

**Special issue statement.**

This article is part of the special issue "GEMS: first year in operation (AMT/ACP inter-journal SI)". It is not associated with
a conference.

**Acknowledgements.**

We thank the NASA GSFC support staff and funding for establishing and maintaining the sites of the PGN used in this investigation. We thank the Principal Investigators (PIs) and staff for their effort in establishing and maintaining the
Seoul_YSU, Seoul_SNU, Seosan, Tsukuba, Ulsan, Yokosuka, Busan, and Bangkok sites. We acknowledge TROPOMI and OMPS science teams for making TROPOMI and OMPS Level 2 data publicly available. We also thank the reviewers and editors for their invaluable comments, which have improved the manuscript.

**Financial support.**

This work was supported by the National Institute of Environmental Research (NIER) of Korea (NIER-2022-04-02-036) and the National Research Foundation of Korea (NRF) grant funded by the Korea government (MSIT) (No. NRF-2020R1C1C1014522).

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
