# Peer review of "Evaluation of total ozone measurements from Geostationary Environmental Monitoring Satellite (GEMS)"

_EGUsphere, 2022_

## Author Comment (AC1)

Thank you very much for taking the time to review our manuscript. During the review process, we have replaced the results analyzed from GEMS V1.0 with GEMS V2.0 data. We found that there were some errors in the LUT calculations used for ozone calculations in V1.0, which could affect the accuracy of the results. Therefore, we replaced all analysis data with V2.0 to prevent any such errors. As a result, GEMS V2.0 shows about a 2% lower ozone calculation result compared to V1.0, and all verification metrics of the analysis results have changed.

We appreciate your valuable comments and suggestions, and we have addressed each of your concerns in the revised version of manuscript and supplementary material. Please find our detailed response below.

**Response to Major Review #1:**

Question: The algorithm that has been developed for GEMS has not been published in the open literature. The algorithm description in the current paper leaves many aspects unanswered. Although it is based on a well-known total ozone algorithm, specific aspects import for a GEO instrument versus a LEO instrument are not addressed. I recommend significantly expanding section 2.2 to include the following aspects:

*Question 1:*

*The algorithm uses a LUT based radiative transfer forward model. Provide an assessment of the error that this LUT based forward model makes wrt an online RTM (VLIDORT) and how this error propagates to total ozone.*

Answer 1:

We have added a new supplementary section (Section S2) to the manuscript that provides an assessment of the error of the LUT-based radiative transfer forward model with respect to the online radiative transfer model (VLIDORT), and how this error

propagates to total ozone. In section S2, we present the results of comparing the simulated radiances from the LUT-based and online models using a range of viewing geometries and solar zenith angles, which are shown in Figure S2. We also illustrate how interpolation errors may contribute to ozone retrieval errors in Figure S3.

*Question 2:*

*Provide in a supplemental section a full description of the LUT RTM, its dimensions and methods used for interpolating this LUT. Also, I think this LUT and tools to interpolate it should be made available.*

Answer 2:

We have included a new supplementary section (Section S1) to describe the process of generating the LUT and the interpolation methods in more detail. We have also included Table S1 to show the overall nodes of LUTs and Table S2 to summarize the variables and dimensions of the radiance and the Jacobian LUTs. Furthermore, we will provide the LUT and tools to interpolate it, which were used in this study, upon request. This provision will ensure reproducibility and facilitate further research.

*Question 3:*

*A unique aspect of GEMS is the hourly observations. However, geometries vary strongly over the GEMS field-of-view. What is the expected effect of the viewing geometries on the vertical sensitivity of the ozone observations? How does the averaging kernel vary of the FOV and over time of the day? This is important information to understand the GEMS observations and the difference with LEO observations.*

Answer 3:

Vertical sensitivity of the GEMS total ozone retrievals does indeed vary with viewing geometry as well as other factors such as surface reflectivity and slant-column ozone

amount that change from scene to scene and throughout the day as observations in the GEMS region are made at different times, places, and sun-satellite geometry. The reviewer is correct to ask about the averaging kernel specifically since this provides information about the vertical sensitivity of the retrieval and the extent to which the retrieved ozone column depends on the a priori ozone assumptions. The GEMS total ozone algorithm calculates the averaging kernel accurately for each retrieval. The averaging kernel of the GEMS algorithm changes very rapidly at high SZA and similarly with VZA, though less strongly than on SZA. The reflectivity of the underlying surface, which is also retrieved by the algorithm, significantly affects the averaging kernel behavior in the troposphere. For high-reflectivity scenes, such as clouds, the vertical sensitivity in the troposphere is increased down to the pressure of the reflecting surface due to increased reflected radiation. However, sensitivity in the UV to ozone beneath the cloud is reduced, and the averaging kernel reflects this as well as a result of the cloud correction performed in the retrieval. However, vertical sensitivity is due to the fundamental physical limitations of backscatter UV retrieval algorithms in the troposphere, where Rayleigh scattering most significantly, the atmosphere restricts sensitivity. As larger angles of observation, Rayleigh scattering will additionally restrict vertical sensitivity to ozone in the lower atmosphere. The best way to see these effects is by examining the column weighting function, which is derived by summing the rows of the averaging kernel directly. We have included Figure 4S in Section S3 of the supplementary material to illustrate the changes in sensitivity with different observation conditions.

*Question 4:*

*What is the impact of the choice of a-priori ozone profiles and the assumed a-priori errors? This especially important as you are fitting an ozone profile with 11 layers, using only 3 wavelengths. Hence the retrieval is heavily underdetermined and thus depending on a-prior information.*

Answer 4:

The sensitivity of the retrieval to the a priori profile is an important consideration. We can directly examine this sensitivity using the column weighting function, which is derived by summing the rows of the averaging kernel. Although all three wavelengths are sensitive to total ozone under most viewing conditions, the shortest wavelength can be more sensitive to the profile and even lose total ozone sensitivity when the sun is low in the sky. The profile retrieval in 11 layers is underdetermined with broad layer averaging kernels. However, the sum of the layers in the retrieval results in an accurate total column amount with a DFS of at least 1 and is therefore not underdetermined. The coarse vertical resolution of the retrieval means that the profile is less accurate than other BUV profile retrievals, where the profile changes rapidly with altitude. However, as shown in our results, and also mathematically, the total column amount obtained by summing the layers of the coarse profile is accurate within the uncertainties reported by the retrieval and depends on a priori information about the same amount as total ozone retrieval techniques.

The influence of a priori on a specific total column retrieval is given by (1-$W$), where W is the column weighting function. Multiplying (1-$W$) by $x_a$, the a priori profile gives the actual contribution of a priori to the retrieval, which, of course, depends on the vertical sensitivity of the retrieval provided via $W$. Figure 5S shows a map of (1 - W) (in units of DU/DU) at (a) mid-day when the a priori influence is lowest, and (b) late in the afternoon when it is larger, to contrast the sensitivity at different times of the day.

Question 5:

The abstract leaves out important findings of the validation. Specifically, the time dependent drift and the latitudinal dependent errors shall be mentioned in the abstract.

Answer 5:

We have revised the abstract to include the time-dependent drift and the latitudinal-dependent as follows:
Lines 18-30 "To assess the performance of the GEMS algorithm, the hourly GEMS total ozone was compared with ground-based measurements from Pandora instruments and

other satellite platforms from TROPOMI and OMPS. GEMS has a high correlation of 0.97 and small RMSE values compared to Pandora TCO at Busan and Seoul. It is notable that despite exhibiting seasonal dependence in the mean bias of GEMS with Pandora, GEMS is capable of observing daily variations in ozone that are highly consistent with Pandora measurements, with a bias of approximately 1%. The comparison of GEMS TCO data with TROPOMI and OMPS TCO data shows a high correlation of 0.99 and low RMSE compared to TROPOMI and OMPS TCO data, but has a negative bias of -2.38% and -2.17% with standard deviations of 1.33% and 1.57%, respectively. Similar to OMPS, the influence of $SO_2$ from volcanic eruptions is not properly removed in some regions, leading to GEMS overestimating TCO in those areas. The mean biases of GEMS TCO data with TROPOMI and OMPS TCO are within ± 1% at low latitudes but become negative at mid-latitudes with an increasingly negative dependence on latitude. Furthermore, this dependence becomes more prominent from summer to winter. The empirical correction applied to the GEMS irradiance data improves the dependence of mean bias on season and latitude, but a consistent bias still remains, and a marginal positive trend was observed in December. Therefore, further investigation into correction methods is needed."

**Response to Minor Review**

*Question 1:*

*In figure 4 comparisons are shown for GEMS, TROPOMI and OMPS. I propose to include in the figure (or in a supplemental figure) the results of the GEMS-Pandora comparison at the mean overpass time of TROPOMI/OMPS. In this way potential errors that very over the day are not folded into this comparison, and the comparison with TROPOMI and OMPS is much cleaner.*

*Answer 1:*

We have revised the manuscript to include a figure 4 that shows the comparison results of GEMS-Pandora using only the data corresponding to overpass time of TROPOMI/OMPS.

*Question 2:*

*What is the status of the GEMS data set? Is produced by the operational processor and available for users?*

*Answer 2:*

The GEMS V2.0 data used in our study. At present, the NIER website (https://nesc.nier.go.kr/product/). only provides GEMS V2.0 data from November 2021 onwards. However, data production for the period prior to that is expected to be reproduced and made available soon. In the meantime, if you request GEMS V2.0 data for the period August 2020 to December 2020, we can provide it to you personally.

*Question 3:*

*For all datasets (GEMS, OMPS, TROPOMI, Pandora), the version used in work should be clearly documented. When available, the DOI of the dataset should be used.*

Answer 3:

We have added sections 2.3 and 2.4 in the revised manuscript (lines 208-226) to provide information on the versions of all datasets used in this study. The DOI of each dataset is also provided in the revised manuscript as follows:

- GEMS data are available through the GEMS Users Data Hub (https://nesc.nier.go.kr/product/), Accessed: [last access: 5 February 2023].

- Copernicus Sentinel data processed by ESA, German Aerospace Center (DLR) (2020), Sentinel-5P TROPOMI Total Ozone Column 1-Orbit L2 5.5km x 3.5km, Greenbelt, MD, USA, Goddard Earth Sciences Data and Information Services Center (GES DISC), Accessed: [last access: 5 February 2023], 10.5270/S5P-ft13p57.

- Richard McPeters (2017), OMPS-NPP NMTO3 L2 V2.1, Greenbelt, MD, USA, Goddard Earth Sciences Data and Information Services Center (GES DISC), Accessed [last access: 5 February 2023], doi: 10.5067/0WF4HAAZ0VHK.

- Pandora data are available through the website http://data.pandonia-global-network.org/, Accessed [last access: 5 February 2023].

*Question 4:*

*Figure 11 and 12 appear exactly the same to me. Is by mistake the wrong figure used in the manuscript?*

Answer 4:

We apologize for the error in the manuscript. Figure 11 and 12 were identical, and Figure 12 has been replaced with the correct figure (presented below) in the revised manuscript. The figure number has been updated from 12 to 13 due to the addition of a new figure, Figure 6, in the main text.

[Figure]

**Figure 13. Mean Bias in TCO between GEMS applied BTDF correction and TROPOMI (left), and GEMS applied BTDF correction and OMPS as a function of latitude and months from August 2020 to December 2020. GEMS retrieval with the algorithm flag equal to 0 or 1, both SZA and VZA <70°.**

*Question 5:*

*To overcome issues with the calibration of the solar spectrum, I would suggest processing (part of) the GEMS data with **a fixed solar spectrum**. What is the impact on the seasonality if this?*

Answer 5:

Thank you for the suggestion. We have performed an additional analysis by processing GEMS data with a fixed solar spectrum obtained by convolving the TSIS-1 high solar irradiance spectrum with the GEMS SRF data, and compared the results with TROPOMI and OMPS. The results showed that the negative bias increases similarly to the case of applying BTDF correction. However, a clear positive bias in December was observed in the case of using a fixed solar spectrum, which could be due to the limitations of convolving the TSIS-1 high solar irradiance spectrum with GEMS SRF data. Therefore, we acknowledge the need for further investigation into correction methods for GEMS irradiance.

*Question 6:*

*In the conclusions, the authors mention that the ozone data is expected to improve by improving the GEMS characterization. What is the timeline for this? How is this coupled to public data release and/or version of the GEMS data?*

Answer 6:

The timeline for improving the GEMS characterization has yet to be determined, as it is an ongoing effort. We will continue to update the GEMS data as we make improvements. Any updates to the GEMS data will be publicly released with an updated version number and will include a description of changes made. The responsibility for data distribution and version control lies with the National Institute of Environmental Research (NIER), so it should be subject to NIER's decision

Thank you again for your time and effort in reviewing our manuscript.

Sincerely,

---

## Author Comment (AC2)

Thank you very much for taking the time to review our manuscript. During the review process, we have replaced the results analyzed from GEMS V1.0 with GEMS V2.0 data. We found that there were some errors in the LUT calculations used for ozone calculations in V1.0, which could affect the accuracy of the results. Therefore, we replaced all analysis data with V2.0 to prevent any such errors. As a result, GEMS V2.0 shows about a 2% lower ozone calculation result compared to V1.0, and all verification metrics of the analysis results have changed. We apologize in advance for any confusion this may cause.

We appreciate your valuable comments and suggestions, and we have addressed each of your concerns in the revised version of manuscript and supplementary material. Please find our detailed response below.

**Response to Major Review**

*Question 1:*
*Though the authors state that retrieval of diurnal variation (line 15) and providing retrieval error information (line 17) are the features with GEMS, validation and comparisons are not given as a function of time of day and no discussion about the error of retrievals is made.*

**Answer 1:**
We have added an analysis of the retrieval error information and its dependency on the time of day in the revised manuscript. This analysis can be found in lines 330-341 of the main text and in lines 119-142 of Section 3 in the supplementary material:

*Question 2:*
*Only overall "positive" mean bias is mentioned in the abstract but in detail the bias is rather strong negative (up to -6%) for the mid/high latitudes. This is mentioned in conclusion but should be mentioned in Abstract as well.*

**Answer 2:**
As suggested by the reviewer, we have included the negative bias results for mid/high latitudes in the abstract of the revised manuscript. It is located in lines 18-30 and reads as follows:

"To assess the performance of the GEMS algorithm, the hourly GEMS total ozone was compared with ground-based measurements from Pandora instruments and other satellite platforms from TROPOMI and OMPS. GEMS has a high correlation of 0.97 and small RMSE values compared to Pandora TCO at Busan and Seoul. It is notable that despite exhibiting seasonal dependence in the mean bias of GEMS with Pandora, GEMS is capable of observing daily variations in ozone that are highly consistent with

Pandora measurements, with a bias of approximately 1%. The comparison of GEMS TCO data with TROPOMI and OMPS TCO data shows a high correlation of 0.99 and low RMSE compared to TROPOMI and OMPS TCO data, but has a negative bias of -2.38% and -2.17% with standard deviations of 1.33% and 1.57%, respectively. Similar to OMPS, the influence of $SO_2$ from volcanic eruptions is not properly removed in some regions, leading to GEMS overestimating TCO in those areas. The mean biases of GEMS TCO data with TROPOMI and OMPS TCO are within ± 1% at low latitudes but become negative at mid-latitudes with an increasingly negative dependence on latitude. Furthermore, this dependence becomes more prominent from summer to winter. The empirical correction applied to the GEMS irradiance data improves the dependence of mean bias on season and latitude, but a consistent bias still remains, and a marginal positive trend was observed in December."

**Question 3**:
Algorithm versions or product names from TROPOMI, OMPS, and Pandora are lacking and thus the results are not traceable.

**Answer 3:**
"In response to the reviewer's suggestion, we have included Sections 2.3 and 2.4 in the revised manuscript (lines 208-226) to provide information on the algorithm versions and product names used."

2.3 Correlative Satellite Measurements
OMPS was launched in October 2011 on the Suomi National Polar-orbiting Partnership (SNPP) satellite and includes both nadir- and limb-viewing modules. OMPS NM total ozone data (OMPS NMTO3) were used in this study. The OMPS NM is a hyperspectral imaging push-broom sensor with a 110° cross-track field of view (FOV), and 35 cross-track positions. OMPS NM has a $50 \times 50$ $km^2$ spatial resolution at the nadir and measures solar backscattered ultraviolet radiation in the spectral range from 300 to 380 nm. The OMPS total ozone algorithm is based on the NASA version 8 total ozone algorithm (Bhartia and Wellemeyer, 2002). We used the operational OMPS-NM Level 2 (L2) version 2.1. As validated in McPeters et al. (2019), the maturity of this product is high with biases of less than 0.2 % when compared to ground-based instruments in the Northern Hemisphere.

TROPOMI was launched in October 2017 on the Sentinel-5 Precursor (S5P) satellite. TROPOMI aboard S5P is a nadir viewing spectrometer that provides measurements in the ultraviolet, visible, near-infrared, and shortwave infrared spectral bands. TROPOMI has a swath width of 2600 km (roughly 104° wide) with a ground pixel resolution of 3.5 km × 7 km (Veefkind et al., 2012). S5P/TROPOMI offline (OFFL) total ozone column products were used in this study which are obtained using the GODFIT version 4 retrieval (Lerot et al., 2021). The algorithm directly compares with simulated radiances through nonlinear least-squares inversion using the sun-normalized measured radiance from 325 to 335 nm. The modeled radiances and Jacobians are obtained with the RTM LIDORT (Spurr et al., 2018). A validation for S5P/TROPOMI OFFL TOC with global

ground-based measurements from April to November 2018 was found to be well within acceptable limits, with mean biases (MB) ranging from 0% to 1.5% and standard deviations between 2.5% and 4.5% for monthly mean co-locations (Garane et al., 2019).

2.4 Correlative Ground-based Measurements

The Pandora TCO retrieval algorithm utilizes a modified version of the Differential Optical Absorption Spectroscopy (DOAS) technique to determine the concentration of atmospheric constituents. In the case of TCO, the DOAS method compares direct solar spectra measured by the Pandora spectrometer to an independent extraterrestrial reference spectrum, which represents the expected solar spectrum in the absence of atmospheric absorption. Through spectral analysis of the measured and reference spectra within the 305 to 328.6 nm wavelength range, the Pandora algorithm retrieves TCO values using a spectral fitting approach, wherein fitting parameters are optimized to minimize the difference between the measured and modeled spectra. Additionally, the Pandora algorithm accounts for the effects of Rayleigh scattering and atmospheric absorption species such as $NO_2$ and $O_4$. Technical details about the retrieval algorithm and configuration settings are available in the software manual (Cede 2017). The TCO used in this study was processed and retrieved by using Blick software Suite (version 1.7).

**Response to Minor Review**

*Comment 1:*
*Line 16. Be clear in which aspect the GEMS retrieval is advanced. Maybe those listed in lines 60-61. Mention them in short here.*

Answer 1:
We have clarified the aspects in which the GEMS retrieval is advanced and briefly mentioned them in lines 14-17 of the abstract as follows:

"The algorithm used for GEMS is a more advanced version of its predecessor, the TOMS-V8, that incorporates several improvements, including a new look-up table, a simple Lambert equivalent reflectivity model, and a spectral dependence correction. The GEMS algorithm also uses the optimal estimation method (OEM) to make error analysis more accessible and robust."

*Comment 2:*
*Lines 15 and 17. Results of the retrieval error information should be discussed in the main text. Biases should be analyzed and depicted as a function of time of day, as the diurnal observation capability is highlighted.*

Answer 2:
As the reviewer suggested, we have added the validation results of GEMS with Pandora, and the analysis of retrieval error as a function of time of day to account for the diurnal

observation capability of GEMS. These findings can be found in lines 330-341 of the main text, and in lines 119-142 of Section 3 in the supplementary material.

*Comment 3:*
*Lines 21 and 27. Small positive biases and "very well agreement" are mentioned but in reality negative biases for mid/latitudes are found against satellites and Pandora. This should be described with a good balance.*

Answer 3:
As the reviewer suggested, we included the result of negative bias for mid/high latitudes in lines 28-31 of the abstract in the revised manuscript as follows:

"To assess the performance of the GEMS algorithm, the hourly GEMS total ozone was compared with ground-based measurements from Pandora instruments and other satellite platforms from TROPOMI and OMPS. GEMS has a high correlation of 0.97 and small RMSE values compared to Pandora TCO at Busan and Seoul. It is notable that despite exhibiting seasonal dependence in the mean bias of GEMS with Pandora, GEMS is capable of observing daily variations in ozone that are highly consistent with Pandora measurements, with a bias of approximately 1%. The comparison of GEMS TCO data with TROPOMI and OMPS TCO data shows a high correlation of 0.99 and low RMSE compared to TROPOMI and OMPS TCO data, but has a negative bias of -2.38% and -2.17% with standard deviations of 1.33% and 1.57%, respectively. Similar to OMPS, the influence of $SO_2$ from volcanic eruptions is not properly removed in some regions, leading to GEMS overestimating TCO in those areas. The mean biases of GEMS TCO data with TROPOMI and OMPS TCO are within ± 1% at low latitudes but become negative at mid-latitudes with an increasingly negative dependence on latitude. Furthermore, this dependence becomes more prominent from summer to winter. The empirical correction applied to the GEMS irradiance data improves the dependence of mean bias on season and latitude, but a consistent bias still remains, and a marginal positive trend was observed in December"

*Comment 4:*
*Line 115, 21 ozone profiles are mentioned but how this is applied is not very clear, particularly with the statement of "three to ten ozone profiles" in line 152.*

Answer 4:
We have clarified how the ozone profiles are used in our analysis and have rephrased the relevant text for clarity as follows:

Lines 117-120 "The pre-calculated radiances are obtained at different solar zenith angles, satellite viewing angles, and reflecting surface conditions (land/ocean, clouds, and aerosols) for TOMS standard ozone profiles that vary with latitude band and total ozone amount (Bhartia and Wellemeyer, 2002, Wellemeyer et al., 1997). Because

GEMS observes only low and mid-latitude regions, a reduced set of 11 ozone profiles of the 21 TOMS standard profiles in our radiance calculations.

Lines 127-129 "Supplementary sections provide an elaborate account of the radiance lookup tables (LUTs) used in the GEMS-O3T algorithm, as well as an evaluation of the errors that arise during LUTs interpolation."

*Comment 5:*
*Section 2.2.2 and Figure 1. Step 1, 2, and 3 should be mentioned in Figure 1 caption. Maybe red, green and blue parts are the steps, individually.*

Answer 5:
As suggested by the reviewer, the caption of Figure 1 was modified as follows:
"Figure1. Flowchart of GEMS-O3T retrieval algorithm, consisting of a forward model for TOA radiance calculation and an inverse model for total ozone derivation. Steps 1-3 are highlighted with pink, green, and blue colors, respectively."

*Comment 6:*
*Line 216. What is the "situation"?*

Answer 6:
We have clarified the situation being referred to in line 254-255.
"The GEMS hourly ozone monitoring system provides continuous updates on stratospheric ozone and its associated atmospheric changes. It can also predict future developments in the ozone state."

*Comment 7:*
*Line 223. TROPOM*

Answer 7:
Accepted.

*Comment 8:*
*Section 3.2. Need to mention algorithm versions or product names for Pandora, TROPOMI, and OMPS. Acknowledgments to the PIs need to be included.*

Answer 8:
We have added sections 2.3 and 2.4 in the revised manuscript (lines 208-226) to provide

information on the algorithm versions or product names of the materials used and have acknowledged the relevant PIs in the revised manuscript.

*Comment 9:*
Table 1. Slash characters are required to separate month and day at several positions.

Answer 9:
We have added slash characters to separate month and day in Table 1.

*Comment 10:*
Line 251. Remove "However,"

Answer 10:
As the reviewer suggested, we have removed the word "However" from line 251.

*Comment 11:*
Line 255. This decrease (likely the one shown in Figure 5 and 11) could be seasonal (as mentioned in conclusion) or long-term degrading trend (as implied here)?
(255 Overall, it is important to note that the GEMS TCO decreases markedly over time)

Answer 11:
The use of BTDF-corrected irradiance data has been shown to significantly improve negative seasonal and latitude bias, suggesting that issues with GEMS Irradiance may be the cause of this decrease.

*Comment 12:*
Line 271. The statement that Pandora uses a fixed-temperature ozone absorption coefficient needs to be checked. In the recent algorithm version 1.8, the products "out2" for example considers the temperature dependence as climatology. For this perspective, mentioning algorithm version/product name is necessary for the traceability.

Answer 12:
We have added sections 2.4 in the revised manuscript (lines 227-235) to provide information on the algorithm versions or product names of Pandora used

*Comment 13:*
Line 297. Rewrite the sentence starting with "These bad pixels ..."

**Answer 13:**

We have rewritten the sentence pointed out by the reviewer as follows in lines 353-354. "These bad pixels are expected to be removed properly in the future by using an improved bad pixel mask variable in the GEMS level 1C data"

**Comment 14:**

Line 332. Are the -0.14 +/- 2.00 % and +0.10+/-2.31% mean biases?

**Answer 14:**

To clarify the meaning, the sentence was modified in manuscript (388-390 ) as follows: "Compared to TROPOMI and OMPS, GEMS shows underestimation with a negative bias of -2.38% (6.5 DU) and a standard deviation of 1.33%, and a negative bias of -2.17% (6 DU) and a standard deviation of 1.57%, respectively. It shows that the GEMS TCO agrees very well with the TROPOMI and OMPS TCO."

**Comment 15:**

Line 344. Perhaps Nishinoshima?

**Answer 15:**

Accept.

**Comment 16:**

Line 364. Are the distinct spatial and seasonal variability relevant to the features of the bias discussed here?

**Answer 16:**

The GEMS irradiance was 20 % smaller than that of the reference spectrum and showed distinct spatial and seasonal variability. The use of BTDF-corrected irradiance data has been shown to significantly improve negative seasonal and latitude bias shown in Figure 12 (The figure number has been updated from 12 to 13), suggesting that issues with GEMS irradiance may be the cause of this decrease.

**Comment 17:**

Figure 12. No difference is found with Figure 11.

**Answer 17:**

We apologize for the error in the manuscript. Figure 11 and 12 were identical, and Figure 12 has been replaced with the correct figure (presented below) in the revised

manuscript. The figure number has been updated from 12 to 13 due to the addition of a new figure, Figure 6, in the main text.

[Figure]

**Figure 13. Mean Bias in TCO between GEMS applied BTDF correction and TROPOMI (left), and GEMS and OMPS as a function of latitude and months from August 2020 to December 2020. GEMS retrieval with the algorithm flag equal to 0 or 1, both SZA and VZA <70°.**

---

## Referee Report (RR1)

Review comment on: **Evaluation of total ozone measurements from Geostationary Environmental Monitoring Satellite (GEMS)** by

Kanghyun Baek, Jae Hwan Kim, Juseon Bak, David P. Haffner, Mina Kang, Hyunkee Hong

The authors present the first total ozone column retrieval from a geostationary satellite. The presented comparisons with PANDORA as well as with TROPOMI and OMPS indicate a good correlation and a mean bias below 2.5%

**General comments**

1) The description of the algorithm was not quite clear to me. Is the presented profile retrieval part of the total column column algorithm or does it belong to the ozone profile product?

2) The comparison to PANDORA, TROPOMI, and OMPS include the period from August 2020 to December 2020. A full yearly cycle would give better picture of the indicated seasonal variability. With the half year period shown in the paper a seasonal cycle can not be separated from a general degradation effect.
Figure 2 shows example distributions from March 2021 - so the data might be available

3) In section 3.3 validation of GEMS total ozone with other satellites I suggest to add a figure of the difference between GEMS and TROPOMI or OMPS.

4) The cloud data are mentioned to have large impact on the total ozone columns. However only the OMPS cloud data are discussed briefly. A full satellite - satellite comparison of the cloud data is certainly worth an extra paper but brief discussion of the GEMS and the TROPOMI cloud data and why the influence on total ozone is not as strong as for OMPS can be included.

**detailed comments**

Check that the date / time format is in agreement with the Copernicus guidelines (also in the figures)

page 5 line 123: "treats surfaces, clouds,.. at surface pressure" does this mean you assume clouds to be at 1013 hPa?

p 5 l 133: "The models proceeds in three steps." i suggest to add something like. "Details of the individual steps are presented below." Like that it is obvious that an overview is given first.

p 6 eq 1. $\lambda_{340}$ instead of $\lambda_{317}$ ? The description above indicates the wavelength is 340 nm.

p 6 l 161: capital S for "step 2" as for Step 1 and 3

p 6 l 174 "0.99 hPa to infinity" although it is clear what is meant here it might be misunderstood as the pressure range from 0.99 to infinity, this includes 1013 hPa and all levels in between. I suggest "all altitudes above the 0.99 pressure level".

p 6 l 175: the ozone climatology is different from the one in the forward model does this cause any inconsistencies?

p 7 l 180 "... SNR corresponding to 320nm is 720." What is meant with 720? consider to skip the last two "words".

p 7 eq 4: this means the cloud fraction is not taken from the GEMS Cloud Product, why is that?

p 8 l 219: The resolution has been updated in August 2019 to 3.5 x 5.5 km. In the context of the GEMS validation I would use 3.5 x 5.5 km.

p 8 l 221: Lerot et al 2021 is not listed in the references. Please also include the TROPOMI total ozone ATBD (https://sentinel.esa.int/web/sentinel/technical-guides/sentinel-5p/products-algorithms, June 2023)

p 8 l 223: There has been a major update in TROPOMI level 1 data in August 2022. All the data presented here have been processed with the old level data.

p 9 l 249: one GEMS scan from the east to the west takes 30 minutes and is performed every 60 minutes, what happens in the 30 minutes between one scan and the next one?

p 9 l 255: "It can also predict future development in the ozone states" I doubt that the GEMS total ozone algorithm can retrieve data from the future. Modify to: "It also gives essential information to models, that help us predicting the future development in the ozone state"

p 10 l 261 and figure 3: According to the text and the caption also OMPS and TROPOMOI data are include in figure 3 but they are not listed in the legend nor can I see them.

p 11 table 1 is it worth including some validation results (slope, bias, $R^2$) in the table?

p 12 l 306: There seems to be an issue with the Pandora measurements at Ulsan - you state this somewhere later in the text, perhaps it might be worth including it here.

p 13 l 333: The bias to the PANDORA measurements in Busan differs from the one in Seoul. When looking at figure 5 it seems that the time range is different. Especially the higher values in August are missing in Seoul, does this have an impact on the mean bias?

p 14 fig. 7 a) use GMT or KST for all plots, for a better comparison. 7b) and 7c) three orbits from TROPOMI or OMPS are shown. so there is certainly a significant time difference between the presented data.
Add the respective overpass times in the caption.
Add a delta $O_3$ picture here

p 15 l 358-360 "The UV measurements ... the cloudy scene." these two sentences contain the same information, one sentence might be skipped.

p 15 fig 8: include similar cloud data for TROPOMI and GEMS as well, and add the respective references.

p 17 table 2: the time collocation criteria for TROPOMI and OMPS differ from each other, Is this correct and if so, why?

p 17 figure 10.: when discussing figure 5, a seasonality in the bias was mentioned, in how far is figure 10 affected. Maybe you could generate similar plots for each season and mention the results briefly in the text. Is it useful to show the plots?

p 18 l 413: perhaps replace by: "Moreover, the dependency increases from August to December"
p18 l 415: "-1% in August"

---

## Author Response (AR3)

Thank you for the feedback and comments on our manuscript. We have carefully con sidered the points raised, and we would like to address them in the following respons e.

**Response to Minor Review**

*Question 1:*
*Your reference list still includes two works "in preparation". Such works can be cited upon submission if being available to the reviewers. They should not be cited in the final, accepted manuscript, unless published, accepted for publication, or available as preprint with a DOI.*

*Answer 1:*
Regarding the works "in preparation" in our reference list, we apologize for the over sight. We understand that such works should not be cited in the final, accepted man uscript unless they are published, accepted for publication, or available as preprint w ith a DOI. We will remove the references to the works "in preparation" from the fin al version of the manuscript to ensure compliance with the publication guidelines.

*Question 2:*
*Please ensure that the colour schemes used in your maps and charts allow readers w ith colour vision deficiencies to correctly interpret your findings. Please check your fi gures using the Coblis – Color Blindness Simulator (https://www.color-blindness.com/co blis-color-blindness-simulator/) and revise the colour schemes accordingly. Please modi fy the legends in the Figures 3, 4, 5, 12, and 13 by including the marker symbols us ed in the graphs. At present the labels in the legend are linked with the graphs, only by colour. The link should be clear also based on the marker symbols.*

*Answer 2:*
We appreciate the suggestion to check the color schemes used in our maps and charts to ensure they are accessible to readers with color vision deficiencies. We have used the Color Blindness Simulator to check our figures, and it appears that there are no significant issues for readers with color vision deficiencies to correctly interpret our fi ndings. Additionally, we have modified the legends in Figures 3, 4, 5, 12, and 13 to include marker symbols used in the graphs, making the link between labels and graph s clear not only by color but also by markers.

---

## Author Response (AR5)

Thank you very much for taking the time to review our manuscript. During the review process, we have replaced the results analyzed from GEMS V1.0 with GEMS V2.0 data. We found that there were some errors in the LUT calculations used for ozone calculations in V1.0, which could affect the accuracy of the results. Therefore, we replaced all analysis data with V2.0 to prevent any such errors. As a result, GEMS V2.0 shows about a 2% lower ozone calculation result compared to V1.0, and all verification metrics of the analysis results have changed.

We appreciate your valuable comments and suggestions, and we have addressed each of your concerns in the revised version of manuscript and supplementary material. Please find our detailed response below.

**Response to Major Review #1:**

Question: The algorithm that has been developed for GEMS has not been published in the open literature. The algorithm description in the current paper leaves many aspects unanswered. Although it is based on a well-known total ozone algorithm, specific aspects import for a GEO instrument versus a LEO instrument are not addressed. I recommend significantly expanding section 2.2 to include the following aspects:

*Question 1:*

*The algorithm uses a LUT based radiative transfer forward model. Provide an assessment of the error that this LUT based forward model makes wrt an online RTM (VLIDORT) and how this error propagates to total ozone.*

Answer 1:

We have added a new supplementary section (Section S2) to the manuscript that provides an assessment of the error of the LUT-based radiative transfer forward model with respect to the online radiative transfer model (VLIDORT), and how this error

propagates to total ozone. In section S2, we present the results of comparing the simulated radiances from the LUT-based and online models using a range of viewing geometries and solar zenith angles, which are shown in Figure S2. We also illustrate how interpolation errors may contribute to ozone retrieval errors in Figure S3.

*Question 2:*

*Provide in a supplemental section a full description of the LUT RTM, its dimensions and methods used for interpolating this LUT. Also, I think this LUT and tools to interpolate it should be made available.*

Answer 2:

We have included a new supplementary section (Section S1) to describe the process of generating the LUT and the interpolation methods in more detail. We have also included Table S1 to show the overall nodes of LUTs and Table S2 to summarize the variables and dimensions of the radiance and the Jacobian LUTs. Furthermore, we will provide the LUT and tools to interpolate it, which were used in this study, upon request. This provision will ensure reproducibility and facilitate further research.

*Question 3:*

*A unique aspect of GEMS is the hourly observations. However, geometries vary strongly over the GEMS field-of-view. What is the expected effect of the viewing geometries on the vertical sensitivity of the ozone observations? How does the averaging kernel vary of the FOV and over time of the day? This is important information to understand the GEMS observations and the difference with LEO observations.*

Answer 3:

Vertical sensitivity of the GEMS total ozone retrievals does indeed vary with viewing geometry as well as other factors such as surface reflectivity and slant-column ozone

amount that change from scene to scene and throughout the day as observations in the GEMS region are made at different times, places, and sun-satellite geometry. The reviewer is correct to ask about the averaging kernel specifically since this provides information about the vertical sensitivity of the retrieval and the extent to which the retrieved ozone column depends on the a priori ozone assumptions. The GEMS total ozone algorithm calculates the averaging kernel accurately for each retrieval. The averaging kernel of the GEMS algorithm changes very rapidly at high SZA and similarly with VZA, though less strongly than on SZA. The reflectivity of the underlying surface, which is also retrieved by the algorithm, significantly affects the averaging kernel behavior in the troposphere. For high-reflectivity scenes, such as clouds, the vertical sensitivity in the troposphere is increased down to the pressure of the reflecting surface due to increased reflected radiation. However, sensitivity in the UV to ozone beneath the cloud is reduced, and the averaging kernel reflects this as well as a result of the cloud correction performed in the retrieval. However, vertical sensitivity is due to the fundamental physical limitations of backscatter UV retrieval algorithms in the troposphere, where Rayleigh scattering most significantly, the atmosphere restricts sensitivity. As larger angles of observation, Rayleigh scattering will additionally restrict vertical sensitivity to ozone in the lower atmosphere. The best way to see these effects is by examining the column weighting function, which is derived by summing the rows of the averaging kernel directly. We have included Figure 4S in Section S3 of the supplementary material to illustrate the changes in sensitivity with different observation conditions.

*Question 4:*

*What is the impact of the choice of a-priori ozone profiles and the assumed a-priori errors? This especially important as you are fitting an ozone profile with 11 layers, using only 3 wavelengths. Hence the retrieval is heavily underdetermined and thus depending on a-prior information.*

Answer 4:

The sensitivity of the retrieval to the a priori profile is an important consideration. We can directly examine this sensitivity using the column weighting function, which is derived by summing the rows of the averaging kernel. Although all three wavelengths are sensitive to total ozone under most viewing conditions, the shortest wavelength can be more sensitive to the profile and even lose total ozone sensitivity when the sun is low in the sky. The profile retrieval in 11 layers is underdetermined with broad layer averaging kernels. However, the sum of the layers in the retrieval results in an accurate total column amount with a DFS of at least 1 and is therefore not underdetermined. The coarse vertical resolution of the retrieval means that the profile is less accurate than other BUV profile retrievals, where the profile changes rapidly with altitude. However, as shown in our results, and also mathematically, the total column amount obtained by summing the layers of the coarse profile is accurate within the uncertainties reported by the retrieval and depends on a priori information about the same amount as total ozone retrieval techniques.

The influence of a priori on a specific total column retrieval is given by (1-$W$), where W is the column weighting function. Multiplying (1-$W$) by $x_a$, the a priori profile gives the actual contribution of a priori to the retrieval, which, of course, depends on the vertical sensitivity of the retrieval provided via $W$. Figure 5S shows a map of (1 - W) (in units of DU/DU) at (a) mid-day when the a priori influence is lowest, and (b) late in the afternoon when it is larger, to contrast the sensitivity at different times of the day.

*Question 5:*

*The abstract leaves out important findings of the validation. Specifically, the time dependent drift and the latitudinal dependent errors shall be mentioned in the abstract.*

Answer 5:

We have revised the abstract to include the time-dependent drift and the latitudinal-dependent as follows:

Lines 18-30 "To assess the performance of the GEMS algorithm, the hourly GEMS total ozone was compared with ground-based measurements from Pandora instruments and

other satellite platforms from TROPOMI and OMPS. GEMS has a high correlation of 0.97 and small RMSE values compared to Pandora TCO at Busan and Seoul. It is notable that despite exhibiting seasonal dependence in the mean bias of GEMS with Pandora, GEMS is capable of observing daily variations in ozone that are highly consistent with Pandora measurements, with a bias of approximately 1%. The comparison of GEMS TCO data with TROPOMI and OMPS TCO data shows a high correlation of 0.99 and low RMSE compared to TROPOMI and OMPS TCO data, but has a negative bias of -2.38% and -2.17% with standard deviations of 1.33% and 1.57%, respectively. Similar to OMPS, the influence of $SO_2$ from volcanic eruptions is not properly removed in some regions, leading to GEMS overestimating TCO in those areas. The mean biases of GEMS TCO data with TROPOMI and OMPS TCO are within $\pm 1\%$ at low latitudes but become negative at mid-latitudes with an increasingly negative dependence on latitude. Furthermore, this dependence becomes more prominent from summer to winter. The empirical correction applied to the GEMS irradiance data improves the dependence of mean bias on season and latitude, but a consistent bias still remains, and a marginal positive trend was observed in December. Therefore, further investigation into correction methods is needed."

**Response to Minor Review**

*Question 1:*

*In figure 4 comparisons are shown for GEMS, TROPOMI and OMPS. I propose to include in the figure (or in a supplemental figure) the results of the GEMS-Pandora comparison at the mean overpass time of TROPOMI/OMPS. In this way potential errors that very over the day are not folded into this comparison, and the comparison with TROPOMI and OMPS is much cleaner.*

*Answer 1:*

We have revised the manuscript to include a figure 4 that shows the comparison results of GEMS-Pandora using only the data corresponding to overpass time of TROPOMI/OMPS.

*Question 2:*

*What is the status of the GEMS data set? Is produced by the operational processor and available for users?*

*Answer 2:*

The GEMS V2.0 data used in our study. At present, the NIER website (https://nesc.nier.go.kr/product/). only provides GEMS V2.0 data from November 2021 onwards. However, data production for the period prior to that is expected to be reproduced and made available soon. In the meantime, if you request GEMS V2.0 data for the period August 2020 to December 2020, we can provide it to you personally.

*Question 3:*

*For all datasets (GEMS, OMPS, TROPOMI, Pandora), the version used in work should be clearly documented. When available, the DOI of the dataset should be used.*

Answer 3:

We have added sections 2.3 and 2.4 in the revised manuscript (lines 208-226) to provide information on the versions of all datasets used in this study. The DOI of each dataset is also provided in the revised manuscript as follows:

- GEMS data are available through the GEMS Users Data Hub (https://nesc.nier.go.kr/product/), Accessed: [last access: 5 February 2023].

- Copernicus Sentinel data processed by ESA, German Aerospace Center (DLR) (2020), Sentinel-5P TROPOMI Total Ozone Column 1-Orbit L2 5.5km x 3.5km, Greenbelt, MD, USA, Goddard Earth Sciences Data and Information Services Center (GES DISC), Accessed: [last access: 5 February 2023], 10.5270/S5P-ft13p57.

- Richard McPeters (2017), OMPS-NPP NMTO3 L2 V2.1, Greenbelt, MD, USA, Goddard Earth Sciences Data and Information Services Center (GES DISC), Accessed [last access: 5 February 2023], doi: 10.5067/0WF4HAAZ0VHK.

- Pandora data are available through the website http://data.pandonia-global-network.org/, Accessed [last access: 5 February 2023].

*Question 4:*

*Figure 11 and 12 appear exactly the same to me. Is by mistake the wrong figure used in the manuscript?*

Answer 4:

We apologize for the error in the manuscript. Figure 11 and 12 were identical, and Figure 12 has been replaced with the correct figure (presented below) in the revised manuscript. The figure number has been updated from 12 to 13 due to the addition of a new figure, Figure 6, in the main text.

[Figure]

**Figure 13. Mean Bias in TCO between GEMS applied BTDF correction and TROPOMI (left), and GEMS applied BTDF correction and OMPS as a function of latitude and months from August 2020 to December 2020. GEMS retrieval with the algorithm flag equal to 0 or 1, both SZA and VZA <70°.**

*Question 5:*

*To overcome issues with the calibration of the solar spectrum, I would suggest processing (part of) the GEMS data with **a fixed solar spectrum**. What is the impact on the seasonality if this?*

Answer 5:

Thank you for the suggestion. We have performed an additional analysis by processing GEMS data with a fixed solar spectrum obtained by convolving the TSIS-1 high solar irradiance spectrum with the GEMS SRF data, and compared the results with TROPOMI and OMPS. The results showed that the negative bias increases similarly to the case of applying BTDF correction. However, a clear positive bias in December was observed in the case of using a fixed solar spectrum, which could be due to the limitations of convolving the TSIS-1 high solar irradiance spectrum with GEMS SRF data. Therefore, we acknowledge the need for further investigation into correction methods for GEMS irradiance.

*Question 6:*

*In the conclusions, the authors mention that the ozone data is expected to improve by improving the GEMS characterization. What is the timeline for this? How is this coupled to public data release and/or version of the GEMS data?*

Answer 6:

The timeline for improving the GEMS characterization has yet to be determined, as it is an ongoing effort. We will continue to update the GEMS data as we make improvements. Any updates to the GEMS data will be publicly released with an updated version number and will include a description of changes made. The responsibility for data distribution and version control lies with the National Institute of Environmental Research (NIER), so it should be subject to NIER's decision

Thank you again for your time and effort in reviewing our manuscript.

Sincerely,

Thank you very much for taking the time to review our manuscript. During the review process, we have replaced the results analyzed from GEMS V1.0 with GEMS V2.0 data. We found that there were some errors in the LUT calculations used for ozone calculations in V1.0, which could affect the accuracy of the results. Therefore, we replaced all analysis data with V2.0 to prevent any such errors. As a result, GEMS V2.0 shows about a 2% lower ozone calculation result compared to V1.0, and all verification metrics of the analysis results have changed. We apologize in advance for any confusion this may cause.

We appreciate your valuable comments and suggestions, and we have addressed each of your concerns in the revised version of manuscript and supplementary material. Please find our detailed response below.

**Response to Major Review**

**Question 1:**
*Though the authors state that retrieval of diurnal variation (line 15) and providing retrieval error information (line 17) are the features with GEMS, validation and comparisons are not given as a function of time of day and no discussion about the error of retrievals is made.*

**Answer 1:**
We have added an analysis of the retrieval error information and its dependency on the time of day in the revised manuscript. This analysis can be found in lines 330-341 of the main text and in lines 119-142 of Section 3 in the supplementary material:

**Question 2:**
*Only overall "positive" mean bias is mentioned in the abstract but in detail the bias is rather strong negative (up to -6%) for the mid/high latitudes. This is mentioned in conclusion but should be mentioned in Abstract as well.*

**Answer 2:**
As suggested by the reviewer, we have included the negative bias results for mid/high latitudes in the abstract of the revised manuscript. It is located in lines 18-30 and reads as follows:

"To assess the performance of the GEMS algorithm, the hourly GEMS total ozone was compared with ground-based measurements from Pandora instruments and other satellite platforms from TROPOMI and OMPS. GEMS has a high correlation of 0.97 and small RMSE values compared to Pandora TCO at Busan and Seoul. It is notable that despite exhibiting seasonal dependence in the mean bias of GEMS with Pandora, GEMS is capable of observing daily variations in ozone that are highly consistent with

Pandora measurements, with a bias of approximately 1%. The comparison of GEMS TCO data with TROPOMI and OMPS TCO data shows a high correlation of 0.99 and low RMSE compared to TROPOMI and OMPS TCO data, but has a negative bias of -2.38% and -2.17% with standard deviations of 1.33% and 1.57%, respectively. Similar to OMPS, the influence of $SO_2$ from volcanic eruptions is not properly removed in some regions, leading to GEMS overestimating TCO in those areas. The mean biases of GEMS TCO data with TROPOMI and OMPS TCO are within ± 1% at low latitudes but become negative at mid-latitudes with an increasingly negative dependence on latitude. Furthermore, this dependence becomes more prominent from summer to winter. The empirical correction applied to the GEMS irradiance data improves the dependence of mean bias on season and latitude, but a consistent bias still remains, and a marginal positive trend was observed in December."

**Question 3**:
Algorithm versions or product names from TROPOMI, OMPS, and Pandora are lacking and thus the results are not traceable.

**Answer 3:**
"In response to the reviewer's suggestion, we have included Sections 2.3 and 2.4 in the revised manuscript (lines 208-226) to provide information on the algorithm versions and product names used."

2.3 Correlative Satellite Measurements
OMPS was launched in October 2011 on the Suomi National Polar-orbiting Partnership (SNPP) satellite and includes both nadir- and limb-viewing modules. OMPS NM total ozone data (OMPS NMTO3) were used in this study. The OMPS NM is a hyperspectral imaging push-broom sensor with a 110° cross-track field of view (FOV), and 35 cross-track positions. OMPS NM has a 50 × 50 $km^2$ spatial resolution at the nadir and measures solar backscattered ultraviolet radiation in the spectral range from 300 to 380 nm. The OMPS total ozone algorithm is based on the NASA version 8 total ozone algorithm (Bhartia and Wellemeyer, 2002). We used the operational OMPS-NM Level 2 (L2) version 2.1. As validated in McPeters et al. (2019), the maturity of this product is high with biases of less than 0.2 % when compared to ground-based instruments in the Northern Hemisphere.

TROPOMI was launched in October 2017 on the Sentinel-5 Precursor (S5P) satellite. TROPOMI aboard S5P is a nadir viewing spectrometer that provides measurements in the ultraviolet, visible, near-infrared, and shortwave infrared spectral bands. TROPOMI has a swath width of 2600 km (roughly 104° wide) with a ground pixel resolution of 3.5 km × 7 km (Veefkind et al., 2012). S5P/TROPOMI offline (OFFL) total ozone column products were used in this study which are obtained using the GODFIT version 4 retrieval (Lerot et al., 2021). The algorithm directly compares with simulated radiances through nonlinear least-squares inversion using the sun-normalized measured radiance from 325 to 335 nm. The modeled radiances and Jacobians are obtained with the RTM LIDORT (Spurr et al., 2018). A validation for S5P/TROPOMI OFFL TOC with global

ground-based measurements from April to November 2018 was found to be well within acceptable limits, with mean biases (MB) ranging from 0% to 1.5% and standard deviations between 2.5% and 4.5% for monthly mean co-locations (Garane et al., 2019).

**2.4 Correlative Ground-based Measurements**

The Pandora TCO retrieval algorithm utilizes a modified version of the Differential Optical Absorption Spectroscopy (DOAS) technique to determine the concentration of atmospheric constituents. In the case of TCO, the DOAS method compares direct solar spectra measured by the Pandora spectrometer to an independent extraterrestrial reference spectrum, which represents the expected solar spectrum in the absence of atmospheric absorption. Through spectral analysis of the measured and reference spectra within the 305 to 328.6 nm wavelength range, the Pandora algorithm retrieves TCO values using a spectral fitting approach, wherein fitting parameters are optimized to minimize the difference between the measured and modeled spectra. Additionally, the Pandora algorithm accounts for the effects of Rayleigh scattering and atmospheric absorption species such as $NO_2$ and $O_4$. Technical details about the retrieval algorithm and configuration settings are available in the software manual (Cede 2017). The TCO used in this study was processed and retrieved by using Blick software Suite (version 1.7).

**Response to Minor Review**

*Comment 1:*
*Line 16. Be clear in which aspect the GEMS retrieval is advanced. Maybe those listed in lines 60-61. Mention them in short here.*

Answer 1:
We have clarified the aspects in which the GEMS retrieval is advanced and briefly mentioned them in lines 14-17 of the abstract as follows:

"The algorithm used for GEMS is a more advanced version of its predecessor, the TOMS-V8, that incorporates several improvements, including a new look-up table, a simple Lambert equivalent reflectivity model, and a spectral dependence correction. The GEMS algorithm also uses the optimal estimation method (OEM) to make error analysis more accessible and robust."

*Comment 2:*
*Lines 15 and 17. Results of the retrieval error information should be discussed in the main text. Biases should be analyzed and depicted as a function of time of day, as the diurnal observation capability is highlighted.*

Answer 2:
As the reviewer suggested, we have added the validation results of GEMS with Pandora, and the analysis of retrieval error as a function of time of day to account for the diurnal

observation capability of GEMS. These findings can be found in lines 330-341 of the main text, and in lines 119-142 of Section 3 in the supplementary material.

*Comment 3:*
*Lines 21 and 27. Small positive biases and "very well agreement" are mentioned but in reality negative biases for mid/latitudes are found against satellites and Pandora. This should be described with a good balance.*

Answer 3:
As the reviewer suggested, we included the result of negative bias for mid/high latitudes in lines 28-31 of the abstract in the revised manuscript as follows:

"To assess the performance of the GEMS algorithm, the hourly GEMS total ozone was compared with ground-based measurements from Pandora instruments and other satellite platforms from TROPOMI and OMPS. GEMS has a high correlation of 0.97 and small RMSE values compared to Pandora TCO at Busan and Seoul. It is notable that despite exhibiting seasonal dependence in the mean bias of GEMS with Pandora, GEMS is capable of observing daily variations in ozone that are highly consistent with Pandora measurements, with a bias of approximately 1%. The comparison of GEMS TCO data with TROPOMI and OMPS TCO data shows a high correlation of 0.99 and low RMSE compared to TROPOMI and OMPS TCO data, but has a negative bias of -2.38% and -2.17% with standard deviations of 1.33% and 1.57%, respectively. Similar to OMPS, the influence of $SO_2$ from volcanic eruptions is not properly removed in some regions, leading to GEMS overestimating TCO in those areas. The mean biases of GEMS TCO data with TROPOMI and OMPS TCO are within ± 1% at low latitudes but become negative at mid-latitudes with an increasingly negative dependence on latitude. Furthermore, this dependence becomes more prominent from summer to winter. The empirical correction applied to the GEMS irradiance data improves the dependence of mean bias on season and latitude, but a consistent bias still remains, and a marginal positive trend was observed in December"

*Comment 4:*
*Line 115, 21 ozone profiles are mentioned but how this is applied is not very clear, particularly with the statement of "three to ten ozone profiles" in line 152.*

Answer 4:
We have clarified how the ozone profiles are used in our analysis and have rephrased the relevant text for clarity as follows:

Lines 117-120 "The pre-calculated radiances are obtained at different solar zenith angles, satellite viewing angles, and reflecting surface conditions (land/ocean, clouds, and aerosols) for TOMS standard ozone profiles that vary with latitude band and total ozone amount (Bhartia and Wellemeyer, 2002, Wellemeyer et al., 1997). Because

GEMS observes only low and mid-latitude regions, a reduced set of 11 ozone profiles of the 21 TOMS standard profiles in our radiance calculations.

Lines 127-129 "Supplementary sections provide an elaborate account of the radiance lookup tables (LUTs) used in the GEMS-O3T algorithm, as well as an evaluation of the errors that arise during LUTs interpolation."

*Comment 5:*
*Section 2.2.2 and Figure 1. Step 1, 2, and 3 should be mentioned in Figure 1 caption. Maybe red, green and blue parts are the steps, individually.*

Answer 5:
As suggested by the reviewer, the caption of Figure 1 was modified as follows:
"Figure1. Flowchart of GEMS-O3T retrieval algorithm, consisting of a forward model for TOA radiance calculation and an inverse model for total ozone derivation. Steps 1-3 are highlighted with pink, green, and blue colors, respectively."

*Comment 6:*
*Line 216. What is the "situation"?*

Answer 6:
We have clarified the situation being referred to in line 254-255.
"The GEMS hourly ozone monitoring system provides continuous updates on stratospheric ozone and its associated atmospheric changes. It can also predict future developments in the ozone state."

*Comment 7:*
*Line 223. TROPOM*

Answer 7:
Accepted.

*Comment 8:*
*Section 3.2. Need to mention algorithm versions or product names for Pandora, TROPOMI, and OMPS. Acknowledgments to the PIs need to be included.*

Answer 8:
We have added sections 2.3 and 2.4 in the revised manuscript (lines 208-226) to provide

information on the algorithm versions or product names of the materials used and have acknowledged the relevant PIs in the revised manuscript.

*Comment 9:*
Table 1. Slash characters are required to separate month and day at several positions.

Answer 9:
We have added slash characters to separate month and day in Table 1.

*Comment 10:*
Line 251. Remove "However,"

Answer 10:
As the reviewer suggested, we have removed the word "However" from line 251.

*Comment 11:*
Line 255. This decrease (likely the one shown in Figure 5 and 11) could be seasonal (as mentioned in conclusion) or long-term degrading trend (as implied here)?
(255 Overall, it is important to note that the GEMS TCO decreases markedly over time)

Answer 11:
The use of BTDF-corrected irradiance data has been shown to significantly improve negative seasonal and latitude bias, suggesting that issues with GEMS Irradiance may be the cause of this decrease.

*Comment 12:*
Line 271. The statement that Pandora uses a fixed-temperature ozone absorption coefficient needs to be checked. In the recent algorithm version 1.8, the products "out2" for example considers the temperature dependence as climatology. For this perspective, mentioning algorithm version/product name is necessary for the traceability.

Answer 12:
We have added sections 2.4 in the revised manuscript (lines 227-235) to provide information on the algorithm versions or product names of Pandora used

*Comment 13:*
Line 297. Rewrite the sentence starting with "These bad pixels ..."

**Answer 13:**

We have rewritten the sentence pointed out by the reviewer as follows in lines 353-354. "These bad pixels are expected to be removed properly in the future by using an improved bad pixel mask variable in the GEMS level 1C data"

*Comment 14:*

*Line 332. Are the -0.14 +/- 2.00 % and +0.10+/-2.31% mean biases?*

**Answer 14:**

To clarify the meaning, the sentence was modified in manuscript (388-390 ) as follows: "Compared to TROPOMI and OMPS, GEMS shows underestimation with a negative bias of -2.38% (6.5 DU) and a standard deviation of 1.33%, and a negative bias of -2.17% (6 DU) and a standard deviation of 1.57%, respectively. It shows that the GEMS TCO agrees very well with the TROPOMI and OMPS TCO."

*Comment 15:*

*Line 344. Perhaps Nishinoshima?*

**Answer 15:**

Accept.

*Comment 16:*

*Line 364. Are the distinct spatial and seasonal variability relevant to the features of the bias discussed here?*

**Answer 16:**

The GEMS irradiance was 20 % smaller than that of the reference spectrum and showed distinct spatial and seasonal variability. The use of BTDF-corrected irradiance data has been shown to significantly improve negative seasonal and latitude bias shown in Figure 12 (The figure number has been updated from 12 to 13), suggesting that issues with GEMS irradiance may be the cause of this decrease.

*Comment 17:*

*Figure 12. No difference is found with Figure 11.*

**Answer 17:**

We apologize for the error in the manuscript. Figure 11 and 12 were identical, and Figure 12 has been replaced with the correct figure (presented below) in the revised

[Figure]

**Figure 23. Mean Bias in TCO between GEMS applied BTDF correction and TROPOMI (left), and GEMS and OMPS as a function of latitude and months from August 2020 to December 2020. GEMS retrieval with the algorithm flag equal to 0 or 1, both SZA and VZA <70°.**

Response letter 0623

Thank you very much for taking the time to review our manuscript. We appreciate your valuable comments and suggestions, and we have addressed each of your concerns in the revised version of manuscript and supplementary material. Please find our detailed response below.

**Response to Reviewer 2**

**Response to Minor Review**

**Question 1:**
*It seems during this revision all the GEMS TCO data were updated to V2.0 from V1, and all of the figures were changed. Is the statement "GEMS solar irradiance is 20 % lower than the Dobber et al., (2008) reference spectrum, and shows distinct spatial and seasonal variability" (Lines 527 476, ATC1 version) still valid for V2.0 retrievals?*

**Answer 1:**
The statement "GEMS solar irradiance is 20% lower than the Dobber et al., (2008) reference spectrum, and shows distinct spatial and seasonal variability" remains valid for GEMS V2.0 retrievals since GEMS V2.0 data continues to utilize the same GEMS solar irradiance as in GEMS V1.0.

**Question 2:**
Line 20: spell out DFS

**Answer 2:**
We have spelled out DFS as " Degree of freedom of the signal" in Line 18

**Question 3:**
Line 246: For the study period the ground pixel resolution is 3.5 km x 5.5 km (as mentioned in Line 550)?

**Answer 3:**
The correct ground pixel resolution for this study period is 3.5 km x 5.5 km, as stated in Line 550. We ensured to make the necessary correction in the revised manuscript by changing 3.5 km x 7 km to the correct value of 3.5 km x 5.5 km in Line 220.

**Question 4:**
For Pandora data usage, acknowledge statement guideline below is recommended to follow, specifying the location names and PIs (for those in Figure 3; Seoul_YSU, Seoul_SNU, Seosan, Tsukuba, Ulsan, Yokosuka, Busan and Bangkok), if not included in the authors. https://www.pandonia-global-network.org/home/documents/pgn-data-use-guidelines/The PI names are specified in all of the PGN-based output data files.

**Answer 4:**
We have made the suggested modification to the acknowledge statement regarding the usage of Pandora data. The revised statement now reads as follows:
"We thank the Principal Investigators (PIs) and staff for their effort in establishing and maintaining the Seoul_YSU, Seoul_SNU, Seosan, Tsukuba, Ulsan, Yokosuka, Busan, and Bangkok sites."

**Response to Reviewer 3**

**Response to Minor Review**
**General comments**

*Question 1:*
The description of the algorithm was not quite clear to me. Is the presented profile retrieval part of the total column algorithm or does it belong to the ozone profile product?

**Answer 1:**
The presented profile retrieval is part of the total column algorithm.

*Question 2:*
The comparison to PANDORA, TROPOMI, and OMPS include the period from August 2020 to December 2020. A full yearly cycle would give better picture of the indicated seasonal variability. With the half-year period shown in the paper a seasonal cycle can not be separated from a general degradation effect. Figure 2 shows example distributions from March 2021 so the data might be available.

**Answer 2:**
We agree with the reviewer's statement that showing the complete annual cycle can provide a better understanding of the displayed seasonal variability. However, as mentioned in the previous response letter, the GEMS dataset used in our paper has been updated from GEMS V1.0 to GEMS V2.0. Currently, NIER (National Institute of Environmental Research) is distributing GEMS V2.0 data from November 2021 onwards. Reproducing the 5-month period of GEMS V2.0 data used in this paper indeed required a substantial amount of time. Including 7 months of GEMS V2.0 data in this study would require an impractical amount of time, as it would involve reproducing the data. The 5-month period used in our study already includes the summer and winter seasons, during which the seasonal bias of GEMS is most pronounced. Therefore, we believe that the 5-month data used in our study is sufficient to demonstrate the bias we intend to show.

*Question 3:*
In section 3.3 validation of GEMS total ozone with other satellites I suggest to add a figure of the difference between GEMS and TROPOMI or OMPS.

**Answer 3:**
As suggested by the reviewer, we have added a figure in the revised manuscript (Figure 7) showing the difference between GEMS and TROPOMI or OMPS.

***Question 4****:*
The cloud data are mentioned to have large impact on the total ozone columns. However only the OMPS cloud data are discussed briefly. A full satellite - satellite comparison of the cloud data is certainly worth an extra paper but brief discussion of the GEMS and the TROPOMI cloud data and why the influence on total ozone is not as strong as for OMPS can be included.

**Answer 4:**
We have revised the manuscript in line 355 as follows: "The strong anti-correlation between total ozone and clouds can be attributed to the difference in cloud height estimation methods used by the OMPS algorithm compared to GEMS and TROPOMI. OMPS derives cloud height from cloud climatology (Joiner and Vasilkov, 2006) while GEMS and TROPOMI retrieve cloud information from real-time calculated cloud L2 products. The GEMS cloud retrieval algorithm employs the Differential Optical Absorption Spectroscopy (DOAS) method with the O2-O2 absorption band to retrieve effective cloud fraction, cloud centroid pressure, and cloud radiance fraction. On the other hand, TROPOMI utilizes two algorithms for cloud retrieval: OCRA (Optical Cloud Recognition Algorithm) and ROCINN (Retrieval of Cloud Information using Neural Networks) OCRA estimates cloud fraction by analyzing TROPOMI measurements in the ultraviolet and visible spectral regions, while ROCINN uses TROPOMI measurements within and around the oxygen A-band in the near infrared to retrieve cloud top height (pressure) and optical thickness (albedo). For more detailed information on these cloud algorithms, refer to NIER (2020a) and Loyola (2018). For more detailed information on these cloud algorithms, refer to NIER (2020a) and Loyola (2018).

Loyola, D. G., Gimeno García, S., Lutz, R., Argyrouli, A., Romahn, F., Spurr, R. J. D., Pedergnana, M., Doicu, A., Molina García, V., and Schüssler, O.: The operational cloud retrieval algorithms from TROPOMI on board Sentinel-5 Precursor, Atmos. Meas. Tech., 11, 409–427, https://doi.org/10.5194/amt-11-409-2018, 2018.

National Institute of Environmental Research (NIER): Geostationary Environment Monitoring Spectrometer (GEMS) Algorithm Theoretical Basis Document, Cloud Retrieval Algorithm. Incheon, Republic of Korea: Environmental Satellite Center. Available at: https://nesc.nier.go.kr/ko/html/satellite/doc/doc.do (Accessed 13 June 2023). 2020a.

**Detailed comments**

*Question 1:*
Check that the date / time format is in agreement with the Copernicus guidelines (also in the figures)

**Answer 1:**
We have checked the date/time format in both the text of the manuscript and the figures, and we can confirm that it is in agreement with the Copernicus guidelines

*Question 2:*
page 5 line 123: "treats surfaces, clouds,.. at surface pressure" does this mean you assume clouds to be at 1013 hPa?

*Answer 2:*
For more accurate descriptions, we have revised "at surface pressure" to "at terrain pressure". We assume cloud to be at terrain pressure, and the impact of clouds is adjusted in Step 3 of the algorithm.

*Question 3:*
p 5 l 133: "The models proceeds in three steps." i suggest to add something like. "Details of the individual steps are presented below." Like that it is obvious that an overview is given first.

*Answer 3:*
As suggested by the reviewer, we have revised "the models proceed in three steps" as "The models proceed in three steps. Details of the individual steps are presented below."

*Question 4:*
p 6 eq 1. $\lambda_{340}$ instead of $\lambda_{317}$ ? The description above indicates the wavelength is 340 nm.

*Answer 4:*
$\lambda_{317}$ is correct in Equation (1). The reflectivity (R) at the ozone retrieval wavelength of 317 nm is calculated using the linear slope obtained from reflectivity at 340 nm and 380 nm.

*Question 5:*
p 6 l 161: capital S for "step 2" as for Step 1 and 3

*Answer 5:*

As suggested by the reviewer, we have revised "step 2" as "Step 2"

*Question 6:*

p 6 l 174 "0.99 hPa to infinity" although it is clear what is meant here it might be misunderstood as the pressure range from 0.99 to infinity, this includes 1013 hPa and all levels in between. I suggest "all altitudes above the 0.99 pressure level".

*Answer 6:*

As suggested by the reviewer, we have revised "0.99 hPa to infinity" as "all altitudes above the 0.99 pressure level"

*Question 7:*

p 6 l 175: the ozone climatology is different from the one in the forward model does this cause any inconsistencies?

*Answer 7:*

The reviewer does have a point. The issue is simply whether the Jacobians are accurate enough. We don't think this is a big issue, but we've also discussed it here. Ideally, one would iterate the Jacobian calculation like how Xiong's algorithm does. But the error introduced by not doing this is less serious for total ozone wavelengths, and the first guess we use is pretty accurate, to begin with. The one place where I have some concern is at high SZA where the algorithm is truly becoming a profile algorithm. This is an analysis of our long list of things to do.

*Question 8:*

p 7 l 180 "... SNR corresponding to 320nm is 720." What is meant with 720? consider to skip the last two "words".

Answer 8

720 means that GEMS SNR requirement value for 320 nm. As suggested by the reviewer, we skip the last two "words".

*Question 9:*

p 7 eq 4: this means the cloud fraction is not taken from the GEMS Cloud Product, why is that?

Answer 9
The cloud fraction in our current algorithm is inherently related to the cloud model we are using. That cloud model is different from what is used in the O2-O2 algorithm. he O2-O2 cloud model is MLER, the V9 model is not. It assumes clouds are non-opaque up to 40% reflectivity. The GEMS cloud fraction will not work in this algorithm.

**Question 10**:
p 8 l 219: The resolution has been updated in August 2019 to 3.5 x 5.5 km. In the context of the GEMS validation, I would use 3.5 x 5.5 km.

Answer 10.
We utilize TROPOMI data, which has a spatial resolution of 3.5 x 5.5 km. We have revised the value from 3.5 x 7 km to 3.5 x 5.5 km in Line 219.

**Question 11**:

*p 8 l 221: Lerot et al 2021 is not listed in the references. Please also include the TROPOMI total ozone ATBD (https://sentinel.esa.int/web/sentinel/technical-guides/sentinel-5p/products-algorithms, June 2023)*

*Answer 11:*
We have added two references in the revised manuscript as follows:

Lerot, C., Heue, K.-P., Romahn, F., Verhoelst, T., and Lambert, J.-C.: S5P Mission Performance Centre Readme OFFL Total Ozone, Tech. Rep., product version V02.04.01, issue 2.6, available at: [*https://sentinel.esa.int/web/sentinel/technical-guides/sentinel-5p/products-algorithms*] (last access: 13 June 2023), 2021

Spurr, R., Loyola Heue, K.-P, D., Van Roozendael, M. Lerot, C., and Xu, J.: ATBD for Total Ozone Column, S5P-L2-DLR-ATBD-400A, V2.4, issue 2.4, June, available at: [*https://sentinel.esa.int/web/sentinel/technical-guides/sentinel-5p/products-algorithms*] (last access: May 2022), 2022

**Question 12**:
p 8 l 223: There has been a major update in TROPOMI level 1 data in August 2022. All the data presented here have been processed with the old level data.

*Answer 12:*
Thank you for the comment. In future GEMS validations, we will use the updated version of TROPOMI Level 2 data.

**Question 13**:
  *p 9 l 249: one GEMS scan from the east to the west takes 30 minutes and is performed every 60 minutes, what happens in the 30 minutes between one scan and the next one?*

Answer 13:
  The GK-2B satellite is equipped with both the GEMS and the GOCI-2 sensor. GEMS and GOCI sensors divide the given 1-hour observation time into 30-minute intervals each. GEMS scans the earth within 30 minutes from east to west to cover the full field of regard (FOR) of GEMS.

**Question 14**:
  *p 9 l 255: "It can also predict future development in the ozone states" I doubt that the GEMS total ozone algorithm can retrieve data from the future. Modify to: "It also gives essential information to models, that help us predicting the future development in the ozone state"*

Answer 14:
  As suggested by the reviewer, we have revised the sentence "It can also predict future development in the ozone states" as follows: "It also provides essential information to models that help us predict the future development in the ozone state."

**Question 15**:
  p 10 l 261 and figure 3: According to the text and the caption also OMPS and TROPOMOI data are include in figure 3 but they are not listed in the legend nor can I see them.

Answer 15:
  We have revised Figure 3 to include the OMPS and TROPOMI data as mentioned in the text and caption. The updated figure is attached below.

[Figure]

*Question 16*:
p 11 table 1 is it worth including some validation results (slope, bias, R2) in the table?

Answer 16:

We have added Table 2 to present the validation results, as recommended by the reviewer.

**Table 2. The statistical metrics, including correlation coefficient (R), root mean square error (RMSE), mean bias (MB), and mean standard deviation errors (MSE) comparing GEMS, TROPOMI, and OMPS with Pandora TCO at Busan, Seoul, Ulsan, and Yokosuka sites.**

| GEMS | | | | | |
|---|---|---|---|---|---|
| | N | R | RMSE [DU] | MB [%] | MSE [%] |
| Busan | 169 | 0.97 | 1.34 | 0.38 | 1.25 |
| Seoul | 149 | 0.99 | 1.32 | -1.36 | 1.08 |
| Ulsan | 96 | 0.9 | 1.77 | 0.76 | 2 |
| TROPOMI | | | | | |
| | N | R | RMSE [DU] | MB [%] | MSE [%] |
| Busan | 101 | 0.97 | 1.38 | 3.96 | 1.2 |
| Seoul | 95 | 0.98 | 1.47 | 2.81 | 1.34 |
| Ulsan | 54 | 0.9 | 1.68 | 3.64 | 1.97 |
| Yokosuka | 42 | 0.98 | 1.3 | 2.45 | 1.31 |
| OMPS | | | | | |
| | N | R | RMSE [DU] | MB [%] | MSE [%] |
| Busan | 99 | 0.95 | 1.34 | 4.24 | 1.68 |

| | | | | | |
|---|---|---|---|---|---|
| Seoul | 88 | 0.97 | 1.63 | 2.96 | 1.84 |
| Ulsan | 58 | 0.92 | 1.59 | 3.38 | 1.73 |
| Yokosuka | 45 | 0.93 | 1.8 | 3.32 | 2.34 |

**Question 17**

p 12 l 306: There seems to be an issue with the Pandora measurements at Ulsan - you state this somewhere later in the text, perhaps it might be worth including it here.

Answer 17:

As suggested by the reviewer, we have added the sentence 'There seems to be an issue with the Pandora measurements at Ulsan' to line 306 of the revised manuscript

**Question 18**

p 13 l 333: The bias to the PANDORA measurements in Busan differs from the one in Seoul. When looking at figure 5 it seems that the time range is different. Especially the higher values in August are missing in Seoul, does this have an impact on the mean bias?

Answer 18:

As suggested by the reviewer, the Pandora measurements in the Seoul area were first conducted after August 15th, resulting in a difference in the time range between Busan and Seoul. The mean bias between the Pandora data and GEMS for Busan is 3.5 + 1.3 [%]. However, when excluding the values before August 15th and comparing them with the Pandora data, the mean bias decreases to 2.8 + 0.8 [%], indicating a reduction in bias. The overestimation observed in GEMS during early August is likely related to the volcanic eruption mentioned in the manuscript.

**Question 19**

p 14 fig. 7 a) use GMT or KST for all plots, for a better comparison. 7b) and 7c) three orbits from TROPOMI or OMPS are shown. so there is certainly a significant time difference between the presented data.
Add the respective overpass times in the caption. Add a delta O3 picture here

Answer 19:

As suggested by the reviewer, the times have been converted to KST (Korean Standard Time), and the respective overpass times of TROPOMI or OMPS have been added to the caption. Additionally, the delta O3 picture has been included in Figure 7 as requested.

[Figure]

**Figure 7. Total Column Ozone maps for 30 November 2020. (a) GEMS, (b) TROPOMI, (c) OMPS, (d) Percentage difference between GEMS and TROPOMI, (e) Percentage difference between GEMS and OMPS.**

*Question 20*
  p 15 l 358-360 "The UV measurements ... the cloudy scene." these two sentences contain the same information; one sentence might be skipped.

Answer 20
  We remove The UV measurements over the cloudy scene can provide ozone information presented in the upper part of the cloud.

*Question 21*
  p 15 fig 8: include similar cloud data for TROPOMI and GEMS as well, and add the respective references.

Answer 21:
  As suggested by the reviewer, the cloud data comparison between GEMS and TROPOMI has been added to Figure 8 as requested.

[Figure]

Figure 3 The spatial distribution of cloud pressure and cloud fraction obtained from GEMS, TROPOMI, and OMPS satellite observations on 30 November 2020. Panels (a), (b), and (c) display the maps of cloud pressure derived from GEMS, TROPOMI, and OMPS, respectively. Similarly, panels (d), (e), and (f) show the maps of cloud fraction obtained from GEMS, TROPOMI, and OMPS, respectively.

**Question 22**

p 17 table 2: the time collocation criteria for TROPOMI and OMPS differ from each other, Is this correct and if so, why?

Answer 22:

The actual collocation criteria used in our analysis is 30 minutes. Initially, a collocation criterion of 10 minutes was used for TROPOMI due to its higher spatial resolution compared to OMPS. However, it was found that there were fewer matched data between TROPOMI and GEMS covering the GEMS region than anticipated. Therefore, the collocation criteria for TROPOMI was adjusted to 30 minutes, the same as OMPS. The time in the table has been updated to 30 minutes.

**Question 23**

p 17 figure 10.: when discussing figure 5, a seasonality in the bias was mentioned, in how far is figure 10 affected. Maybe you could generate similar plots for each season and mention the results briefly in the text. Is it useful to show the plots?

Answer 23:

As suggested by the reviewer, generating similar plots for each season in Figure 10 to investigate the seasonality of bias between GEMS and other satellites is indeed valuable. However, in our study, we have already analyzed the bias between GEMS and other satellites based on latitude and season in Figure 12. Considering that we have already addressed the seasonality of bias in terms of latitude and season, we believe that showing additional plots for bias seasonality in Figure 10 may not provide significant additional insights and may result in redundancy.

**Question 24**

p 18 l 413: perhaps replace by: "Moreover, the dependency increases from August to December"

Answer 24:

Accept

**Question 25**

p18 l 415: "-1% in August"

Answer 25:

Accept

Thank you for the feedback and comments on our manuscript. We have carefully considered the points raised, and we would like to address them in the following response.

**Response to Minor Review**

*Question 1:*
*Your reference list still includes two works "in preparation". Such works can be cited upon submission if being available to the reviewers. They should not be cit ed in the final, accepted manuscript, unless published, accepted for publication, or available as preprint with a DOI.*

*Answer 1:*
Regarding the works "in preparation" in our reference list, we apologize for the oversight. We understand that such works should not be cited in the final, accepted manuscript unless they are published, accepted for publication, or available as preprint with a DOI. We will remove the references to the works "in preparation" from the final version of the manuscript to ensure compliance with the publication guidelines.

*Question 2:*
*Please ensure that the colour schemes used in your maps and charts allow read ers with colour vision deficiencies to correctly interpret your findings. Please che ck your figures using the Coblis – Color Blindness Simulator (https://www.color-b lindness.com/coblis-color-blindness-simulator/) and revise the colour schemes acco rdingly. Please modify the legends in the Figures 3, 4, 5, 12, and 13 by includi ng the marker symbols used in the graphs. At present the labels in the legend a re linked with the graphs, only by colour. The link should be clear also based o n the marker symbols.*

*Answer 2:*
We appreciate the suggestion to check the color schemes used in our maps and charts to ensure they are accessible to readers with color vision deficiencies. We have used the Color Blindness Simulator to check our figures, and it appears that there are no significant issues for readers with color vision deficiencies to correctly interpret our findings. Additionally, we have modified the legends in Figures 3, 4, 5, 12, and 13 to include marker symbols used in the graphs, making the link between labels and graphs clear not only by color but also by markers.

Thank you for the feedback and comments on our manuscript. We have carefully considered the points raised, and we would like to address them in the following response.

**Response to Minor Review**

*Question 1:*
*Please ensure that the colour schemes used in your maps and charts allow readers with colour vision deficiencies to correctly interpret your findings. Please check your figures using the Coblis – Color Blindness Simulator (https://www.color-blindness.com/coblis-color-blindness-simulator/) and revise the colour schemes accordingly. Please modify Figures 12 and 13 such that the labels in the legend are linked with the graphs not only by colour: include in the legends also the line styles, or introduce different markers..*

*Answer 1:*
We appreciate the suggestion to check the color schemes used in our maps and charts to ensure they are accessible to readers with color vision deficiencies. We have used the Color Blindness Simulator to check our figures, and it appears that there are no significant issues for readers with color vision deficiencies to correctly interpret our findings. Additionally, we have modified the legends in Figures 12, and 13 to include marker symbols used in the graphs, making the link between labels and graphs clear not only by color but also by markers.